# Learning Robust Neural Processes with Risk-Averse Stochastic Optimization

**Huafeng Liu** [1 2]   **Yiran Fu** [1 2]   **Liping Jing** [1 2 3]   **Hui Li** [1 2]   **Shuyang Lin** [1 2]
**Jingyue Shi** [1 2]   **Deqiang Ouyang** [4]   **Jian Yu** [1 2]

## Abstract

Neural processes (NPs) are a promising paradigm to enable skill transfer learning across tasks with the aid of the distribution of functions. The previous NPs employ the empirical risk minimization principle in optimization. However, the fast adaption ability to different tasks can vary widely, and the worst fast adaptation can be catastrophic in risk-sensitive tasks. To achieve robust neural processes modeling, we consider the problem of training models in a risk-averse manner, which can control the worst fast adaption cases at a certain probabilistic level. By transferring the risk minimization problem to a two-level finite sum minimax optimization problem, we can easily solve it via a double-looped stochastic mirror prox algorithm with a task-aware variance reduction mechanism via sampling samples across all tasks. The mirror prox technique ensures better handling of complex constraint sets and non-Euclidean geometries, making the optimization adaptable to various tasks. The final solution, by aggregating prox points with the adaptive learning rates, enables a stable and high-quality output. The proposed learning strategy can work with various NPs flexibly and achieves less biased approximation with a theoretical guarantee. To illustrate the superiority of the proposed model, we perform experiments on both synthetic and real-world data, and the results demonstrate that our approach not only helps to achieve more accurate performance but also improves model robustness.

[1]Beijing Key Laboratory of Traffic Data Mining and Embodied Intelligence, Beijing, China [2]School of Computer Science and Technology, Beijing Jiaotong University, Beijing, China [3]State Key Laboratory of Advanced Rail Autonomous Operation, Beijing, China [4]Collage of Computer Science, Chongqing University, Chongqing, China. Correspondence to: Deqiang Ouyang, Liping Jing <deqiangouyang@cqu.edu.cn, lpjing@bjtu.edu.cn>.

*Proceedings of the $42^{nd}$ International Conference on Machine Learning*, Vancouver, Canada. PMLR 267, 2025. Copyright 2025 by the author(s).

## 1. Introduction

Neural processes (NPs) (Garnelo et al., 2018a;b) represent a class of approximation models for Gaussian processes (GPs) (Rasmussen, 2003), offering promising attributes in terms of computational efficiency and uncertainty quantification. Unlike conventional statistical modeling, where a user typically needs to manually specify a prior, such as the smoothness of functions characterized by a Gaussian distribution in Gaussian processes, NPs subtly define a broad spectrum of stochastic processes using neural networks in a data-driven fashion. When trained appropriately, NPs can delineate a flexible range of stochastic processes that are particularly apt for representing complex functions that existing stochastic processes struggle to capture.

NPs model the relationship between inputs and outputs in the form of probability distributions. This allows the model to generate predictions with associated uncertainty, which is particularly beneficial when only a small amount of training data is available. By capturing the model's confidence, NPs are more robust in few-shot scenarios. NPs can leverage meta-learning to extract shared knowledge from multiple related tasks. This enables the model to quickly adapt to new tasks by utilizing the information learned from previous tasks. Even with limited data in the current task, the model can make accurate predictions by drawing on this shared knowledge. Unlike traditional Bayesian methods, NPs do not depend on explicit prior assumptions. Instead, they learn directly from the data, making them more flexible and effective in few-shot learning contexts. Thanks to the characteristics above of NPs, many researchers are focusing on the few-shot learning capabilities of NPs (Gordon et al., 2019; Liu et al., 2024b; Bruinsma et al., 2023).

However, the optimization in most of the previous NPs optimizations has adopted the empirical risk minimization principle, overlooking the differences in rapid adaptation between tasks. Given a batch of training tasks, NPs treat the task weights uniformly, which can result in insufficient training for the fast adaptation capability to certain tasks. As shown in Figure 1, we present the original task risk frequency statistics. It can be observed that high-risk tasks (with risk values above 70%) account for more than 30% of the total tasks. Therefore, when a new task matches

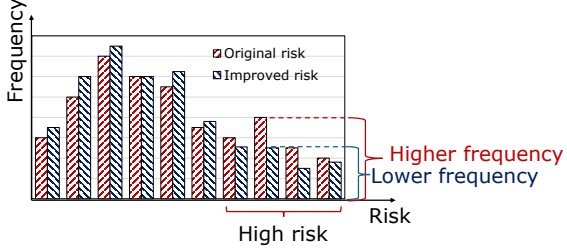

*Figure 1.* Illustration of the task risk distributions of original NPs and our robust NPs.

these high-risk tail tasks, the prediction risk for the new task is similarly high, thereby affecting the performance of the new task. This issue is particularly critical in sensitive scenarios, such as spike sorting (Pakman et al., 2020), and robot navigation (Yildirim & Ugur, 2022).

In this paper, we focus on the performance difference in fast adaptation to various tasks as an indispensable consideration. By investigating the learning paradigm from the view of risk distribution, we convert the optimization principles for NPs to stochastic optimization problems while minimizing the expected tail risk, i.e., conditional value-at-risk (CVaR) (Rockafellar et al., 2000). Specifically, we focus on expected tail risk minimization for NPs with CVaR, which can reshape the task risk distribution and control the task's fast adaptation ability at a certain probabilistic level. When using a variant of SGD for solving the tail risk, it is common to approximate the expectation through mini-batch sampling. However, if the batch is sampled uniformly at random, only a small fraction (e.g., $\alpha$ in CVaR) of the selected samples may carry meaningful gradient information. For the remaining points, their gradients are often truncated to zero due to the presence of the $\max \cdot$ operation and the inherent non-linearity. On the other hand, for the points that do contribute, their gradients are amplified by a factor of a small number (e.g., $1/\alpha$), which can result in exploding gradients. Meanwhile, only a small number of higi-loss samples contribute to the gradient, and the variance of the gradient estimates is significantly high, which makes the optimization trajectory highly unstable. These combined issues make stochastic optimization highly challenging.

Inspired by the previous work (Alacaoglu & Malitsky, 2022), we formulate the CVaR problem as a distributionally robust optimization problem with stochastic minimax optimization and introduce a variance-reduced stochastic mirror prox algorithm. Specifically, we follow the double-loop structure of variance reduction and leverage the two-level structure to compute the stochastic gradient based on a sampling strategy across all tasks, which can get a variance-reduced stochastic gradient estimator. The final solution, by aggregating prox points with the adaptive learning rates, enables a stable and high-quality output. The proposed risk

and optimization algorithm can be flexible to work with various NPs and achieve a less biased approximation. Extensive experiments conducted on both synthetic and real-world datasets demonstrate that the proposed solution can improve robustness.

## 2. Related Work

In this section, we briefly review three different areas that are highly relevant to the proposed method: neural processes and risk-sensitive optimization.

**Neural Processes** Neural Processes (NPs) are a class of models that aim to combine the flexibility of neural networks with the probabilistic framework of Gaussian Processes (GPs). Introduced by (Garnelo et al., 2018a), NPs learn a distribution over functions by conditioning on a set of context points. This allows them to perform tasks such as few-shot learning and function approximation with remarkable efficiency. Several mechanisms or techniques have been integrated into NPs to enhance their performance and applicability, such as attention mechanisms (Kim et al., 2019), hierarchical priors (Wang & Van Hoof, 2020), translation equivariance (Gordon et al., 2019; Kawano et al., 2021), bootstraping (Lee et al., 2020). In recent years, cutting-edge technologies have been applied to NPs to enhance model performance, such as neural ODE (Norcliffe et al., 2021), transformer (Nguyen & Grover, 2022; Maraval et al., 2024), diffusion model (Dutordoir et al., 2023), contrastive learning (Ye & Yao, 2022), autoregressive mechanism (Bruinsma et al., 2023; Tai, 2023). Unlike making improvements to the model, many researchers have also explored the applicability of NPs to different specific tasks, for example, recommender systems (Lin et al., 2021; Liu et al., 2022; 2024b), hyperparameter optimization (Wei et al., 2021), neuroscience (Cotton et al., 2020; Pakman et al., 2020), space science (Park & Choi, 2021), and physics-informed systems design (Vadeboncoeur et al., 2023).

**Risk-sensitive Optimization** Robust Optimization (RO) (Gabrel et al., 2014; Rahimian & Mehrotra, 2019; Yu et al., 2024) aims to improve model stability and reliability by designing algorithms capable of handling uncertainty and perturbations. The core idea of robust optimization is to consider the worst-case performance of a model with min-max formulation during the training process, rather than just its expected performance (Bertsimas & Sim, 2004) by introducing uncertainty sets or adversarial perturbations. Robust optimization strategies typically include two major categories: adversarial robustness and distributionally robust optimization. Adversarial examples are a specific form of perturbation in machine learning, where an attacker can deceive the model by making small perturbations to the input (adversarial attacks) (Gabrel et al., 2014; ?). Distributionally robust optimization is a

method that focuses on ensuring model robustness when the training data distribution changes(Rahimian & Mehrotra, 2019). Among them, the risk principle Value-at-Risk (VaR), as a representative method of robust optimization with probabilistic constraints, was first introduced in the financial sector (Rockafellar et al., 2000), gained significant attention (Quaranta & Zaffaroni, 2008), and has since been widely applied to other fields, such as healthcare (Chen et al., 2019), reinforcement learning (Pinto et al., 2017; Gagne & Dayan, 2021), smart grid (Ben-Tal & Nemirovski, 1999; Bertsimas et al., 2012), and so on. As for robust strategies for NPs, some researchers have explored approaches such as using bootstrapping (Lee et al., 2020) or ensemble methods (Liu et al., 2024a).

# 3. Preliminaries

Let calligraphic letter (e.g., $\mathcal{A}$) indicate set, capital letter (e.g., $A$) for scalar, lower-case bold letter (e.g., $\mathbf{a}$) for vector, and capital bold letter (e.g., $\mathbf{A}$) for matrix. Suppose there is a dataset $\mathcal{D} = (\mathbf{X}, \mathbf{y}) = \{(\mathbf{x}_i, y_i)\}_{i=1}^{N}$ with $N$ data points $\mathbf{X} = [\mathbf{x}_1, \mathbf{x}_2, \cdots, \mathbf{x}_N]^{\top} \in \mathbb{R}^{N \times D}$, and corresponding labels $\mathbf{y} = [y_1, y_2, \cdots, y_N] \in \mathbb{R}^{N}$. Considering an arbitrary number of data points $\mathcal{D}^{\mathcal{C}} = (\mathbf{X}^{\mathcal{C}}, \mathbf{y}^{\mathcal{C}}) = \{(\mathbf{x}_i, y_i)\}_{i \in \mathcal{C}}$, where $\mathcal{C} \subseteq \{1, 2, \cdots, N\}$ is an index set defining context information, neural processes model the conditional distribution of the target values $\mathbf{y}^{\mathcal{T}} = \{y_i\}_{i \in \mathcal{T}}$ at some target data points $\mathbf{X}^{\mathcal{T}} = \{\mathbf{x}_i\}_{i \in \mathcal{T}}$ based on the context $\mathcal{D}^{\mathcal{C}}$, i.e., $\mathbb{E}_{p(\mathcal{C})} p_{\theta}(\mathbf{y}^{\mathcal{T}} | \mathbf{X}^{\mathcal{T}}, \mathcal{D}^{\mathcal{C}}) = \mathbb{E}_{p(\mathcal{C})} \sum_{i \in \mathcal{T}} p_{\theta}(y_i | \mathbf{x}_i, \mathcal{D}^{\mathcal{C}})$, where $p(\mathcal{C})$ is a context prior indicating selecting context information from the whole data set $\mathcal{D}$, e.g., $|\mathcal{C}| \sim \mathrm{U}[1, \cdots, N]$. Usually, target set is defined as $\mathcal{T} = \{1, 2, \cdots, N\}$. Only in CNP (Garnelo et al., 2018a), $\mathcal{T} \subseteq \{1, 2, \cdots, N\}$ and $\mathcal{T} \cap \mathcal{C} = \emptyset$.

Usually, a latent variable $\mathbf{z}$ is introduced to capture model uncertainty and the NPs infer $q_{\theta}(\mathbf{z} | \mathcal{D}^{\mathcal{C}})$ given context set using the reparameterization trick (Kingma & Welling, 2013) and models such a conditional distribution as $p_{\theta}(y_i | \mathbf{x}_i, \mathcal{D}^{\mathcal{C}}) = \int p_{\theta}(y_i | \mathbf{x}_i, \mathcal{D}^{\mathcal{C}}, \mathbf{z}) q_{\theta}(\mathbf{z} | \mathcal{D}^{\mathcal{C}}) d\mathbf{z}$, and it is trained by maximizing an ELBO: $\ell(\mathcal{D}^{\mathcal{C}}, \mathcal{D}^{\mathcal{T}}; \theta) = \mathbb{E}_{\mathbf{z} \sim q_{\theta}(\mathbf{z}|\mathcal{D}^{\mathcal{T}})}[\log p_{\theta}(\mathbf{y}^{\mathcal{T}} | \mathbf{x}^{\mathcal{T}})] - KL[q_{\theta}(\mathbf{z}|\mathcal{D}^{\mathcal{T}}) \| p_{\theta}(\mathbf{z}|\mathcal{D}^{\mathcal{C}})]$. The problems are difficult non-convex optimization problems, especially when the neural networks are introduced, and iterative methods, e.g., gradient-based methods, are adopted to find solutions converging to local minimum (Wang & Van Hoof, 2020).

## 3.1. Risk Minimization of NPs

To achieve robust NPs and inspired by (Wang et al., 2024), we can optimize CVaR of the task risk function for NPs with the adjustable confidence level $\alpha$, and define controllable expected tail task risk based on the CVaR of the task risk function.

To achieve fast prediction on a new context set at test time, NPs *meta-learn* a distribution over predictors. Consider the distribution of tasks $p(\tau)$ for NPs. Let the task space be $\Omega_{\tau}$ and $\tau$ be the task sampled from the distribution $p(\tau)$. To perform meta-learning, we require a meta-dataset (dataset of datasets) $\mathcal{D}_{\tau, 1:M} = \{\mathcal{D}_{\tau, m}\}_{m=1}^{M}$ with contains $M$ datasets. In NPs, the task risk function $\ell(\mathcal{D}_{\tau}^{\mathcal{C}}, \mathcal{D}_{\tau}^{\mathcal{T}}; \theta)$ evaluetes the model's adaption performance. Based on the commonly used risk minimization principle, the expected risk minimization of NPs is given as follows.

**Definition 3.1.** (Expected Risk Minimization for NPs) Given the task distribution $p(\tau)$, the expected risk minimization for NPs is given as

$$\min_{\theta \in \Theta} \hat{R}_{\tau}(\theta) := \mathbb{E}_{p(\tau)}[\ell(\mathcal{D}_{\tau}^{\mathcal{C}}, \mathcal{D}_{\tau}^{\mathcal{T}}; \theta)]. \tag{1}$$

The task distribution determines the diversity and breadth of the meta-dataset, which plays a crucial role in fast adaptation. By performing Monte Carlo estimation to Eq.(1), we can get the empirical risk minimization as $\min_{\theta} \frac{1}{B} \sum_{i=1}^{B} [\ell(\mathcal{D}_{\tau, i}^{\mathcal{C}}, \mathcal{D}_{\tau, i}^{\mathcal{T}}; \theta)]$.

Next, we focus on defining the task risk functions as random variables. Given the task space $\Omega_{\tau}$, we can define a probability measure $p : \delta_{\tau} \mapsto [0, 1]$ over the task space as $(\Omega_{\tau}, \delta_{\tau}, p)$, where $\delta_{\tau}$ is a $\sigma$-algebra on the subsets of task space. Let $(\mathbb{R}^{+}, \mathcal{B})$ be a probability measure over the non-negative real domain defined by the task risk function $\ell(\mathcal{D}_{\tau}^{\mathcal{C}}, \mathcal{D}_{\tau}^{\mathcal{T}}; \theta)$ and $\mathcal{B}$ be a Borel $\sigma$-algebra, the NP operator $NP_{\theta} : \Omega_{\tau} \mapsto \mathbb{R}^{+}$ is given as $NP_{\theta} : \tau \mapsto \ell(\mathcal{D}_{\tau}^{\mathcal{C}}, \mathcal{D}_{\tau}^{\mathcal{T}}; \theta)$.

In this case, the task risk function $\ell(\mathcal{D}_{\tau}^{\mathcal{C}}, \mathcal{D}_{\tau}^{\mathcal{T}}; \theta)$ can be viewed as a random variable, and we can induce a distribution $p(\ell(\mathcal{D}_{\tau}^{\mathcal{C}}, \mathcal{D}_{\tau}^{\mathcal{T}}; \theta))$ over the task risk function. The cumulative distribution function (CDF) of a real-valued random variable $\ell(\mathcal{D}_{\tau}^{\mathcal{C}}, \mathcal{D}_{\tau}^{\mathcal{T}}; \theta)$ can be formulated as $F_{\ell}(\xi; \theta) = p(\ell(\mathcal{D}_{\tau}^{\mathcal{C}}, \mathcal{D}_{\tau}^{\mathcal{T}}; \theta) \leq \xi)$, where $\xi \in \mathbb{R}^{+}$. We cannot give an explicit form of this CDF since it implicitly depends on the loss function $\ell(\cdot)$ and model parameter $\theta$.

Based on the definition of VaR and CVar (Eq.(16)), we can define the VaR and CVaR of the task risk function.

**Definition 3.2.** Given the task risk function $\ell(\mathcal{D}_{\tau, 1:M}; \theta)$ with cumulative distribution function $F_{\ell}(\xi; \theta)$, the VaR of the task risk function at level $\alpha \in (0, 1)$ is given as $v_{\alpha}(\ell(\mathcal{D}_{\tau, 1:M}; \theta)) = \inf\{\xi | F_{\ell}(\xi; \theta) \geq \alpha\}$.

**Definition 3.3.** Given the task risk function $\ell(\mathcal{D}_{\tau, 1:M}; \theta)$ with cumulative distribution function $F_{\ell}(\xi; \theta)$ and the VaR of the task risk function $v_{\alpha}(\ell(\mathcal{D}_{\tau, 1:M}; \theta))$. Assume the random variable is constrained by $\ell(\mathcal{D}_{\tau}^{\mathcal{C}}, \mathcal{D}_{\tau}^{\mathcal{T}}; \theta) \geq v_{\alpha}(\ell(\mathcal{D}_{\tau, 1:M}; \theta))$, the CVaR of the task risk function is defined as $c_{\alpha}(\ell(\mathcal{D}_{\tau, 1:M}; \theta)) = \int_{0}^{\infty} \xi dF_{\ell}^{\alpha}(\xi; \theta)$, where $F_{\ell}^{\alpha}(\xi; \theta)$ is the normalized cumulative distribution, which

is

$$F_\ell^\alpha(\xi;\theta) = \begin{cases} 0 & \xi < v_\alpha(\ell(\mathcal{D}_{\tau,1:M};\theta)) \\ \frac{F_\ell(\xi;\theta) - \alpha}{1-\alpha} & \xi \geq v_\alpha(\ell(\mathcal{D}_{\tau,1:M};\theta)) \end{cases} \quad (2)$$

Thus, we can obtain the normalized probability measure $(\Omega_{\alpha,\tau}, \delta_{\alpha,\tau}, p_\alpha)$, where $\Omega_{\alpha,\tau} = \cup_{\xi \geq v_\alpha(\ell(\mathcal{D}_{\tau,1:M};\theta))}[NP_\theta^{-1}]$. In this case, the task distribution constrained in $\Omega_{\alpha,\tau}$ is denoted by $p_\alpha(\tau)$.

### 3.2. Learning with the CVaR

Given the constrained task distribution $p_\alpha(\tau)$, the controllable expected tail task risk is defined as minimizing CVaR $c_\alpha(\ell(\mathcal{D}_{\tau,1:M};\theta))$, which can be rewritten as follows:

$$\min_{\theta \in \Theta} \hat{R}_{\alpha,\tau}(\theta) := \mathbb{E}_{p_\alpha(\tau)}\left[\ell(\mathcal{D}_\tau^\mathcal{C}, \mathcal{D}_\tau^\mathcal{T}; \theta)\right]. \quad (3)$$

However, directly optimizing the above controllable expected tail task risk is intractable since the constrained task distribution $p_\alpha(\tau)$ implicitly depends on $\theta$ and $\alpha$ and cannot access an explicit closed-form expression.

To avoid directly computing the constrained task distribution $p_\alpha(\tau)$, by introducing slack variable $l$ and auxiliary risk function $[\ell(\mathcal{D}_\tau^\mathcal{C}, \mathcal{D}_\tau^\mathcal{T}; \theta) - l]^+$, the probability-constrained function in Eq.(3) can be converted into the following unconstrained form:

$$\min_{\theta \in \Theta, l \in \mathbb{R}} \hat{R}_\alpha(l,\theta) := l + \frac{1}{1-\alpha}\mathbb{E}_{p(\tau)}\left[\left[\ell(\mathcal{D}_\tau^\mathcal{C}, \mathcal{D}_\tau^\mathcal{T}; \theta) - l\right]^+\right]. \quad (4)$$

Let samples from $\mathcal{D}_{\tau,m}$ be $\{Z_{m,i}\}_{i=1}^{n_m}$, where $n_m$ is the number of samples in the dataset $\mathcal{D}_{\tau,m}$ and $Z = (\mathbf{x}, y)$. By replacing the expectation $\mathbb{E}_{p(\tau)}$ with the empirical tasks, the problem (4) yields the following learning problem:

$$\min_{\theta \in \Theta, l \in \mathbb{R}} R_\alpha(l,\theta) := l + \frac{1}{(1-\alpha)M}\sum_{m=1}^{M}\left[[R_m(\theta) - l]^+\right]. \quad (5)$$

where $R_m(\theta) = \frac{1}{n_m}\sum_{j=1}^{n_m} \ell_m(Z_{m,j}; \theta)$. This formulation focuses on the tail distribution of the loss by truncating smaller losses and optimizing only large ones.

### 3.3. Challenges for Stochastic Optimization

A variant of the SGD method can optimize the above problem. However, there are several challenges in CVaR optimization: 1) *Vanishing Gradient Problem*: In the CVaR optimization formula, gradients are non-zero only for samples where the loss $\ell(\mathcal{D}_{\tau,i}^\mathcal{C}, \mathcal{D}_{\tau,i}^\mathcal{T}; \theta)$ exceeds the truncation threshold $s$. However, only a small fraction of samples (proportional to $\alpha$) typically satisfy this condition. As a result, most samples contribute no gradient, leading to fewer effective updates and slowing or even halting the optimization process. 2) *Exploding Gradient Problem*: For samples

where $\ell(\mathcal{D}_{\tau,i}^\mathcal{C}, \mathcal{D}_{\tau,i}^\mathcal{T}; \theta) \geq s$, the gradient is scaled by $1/\alpha$. Since $\alpha$ is usually a small value (e.g., 0.01 or 0.05), this scaling can cause the gradient magnitude to grow excessively large, leading to exploding gradients and destabilizing the optimization process. 3) *High Gradient Variance*: Because only a small number of high-loss samples contribute to the gradient, the variance of the gradient estimates is significantly high. This high variance makes the optimization trajectory highly unstable and can cause the algorithm to converge slowly or get stuck in local extrema. In summary, the combination of vanishing gradients, exploding gradients, and high gradient variance leads to inefficiency and instability in stochastic optimization algorithms for CVaR objectives.

## 4. Algorithm for Tail Task Risk Optimization

The CVaR objective has a natural distributionally robust optimization (DRO) formulation. Given the DRO set $\mathcal{Q}_\alpha = \{q \in \mathbb{R}^M | 1 \leq q_m \leq 1/k, \sum_m q_m = 1\}$ with $k = \lfloor \alpha M \rfloor$, the DRO problem can be formulated as the following min-max problem:

$$\min_{\theta \in \Theta} \max_{q \in \mathcal{Q}_\alpha}\left\{F_\alpha(\theta, q) := \sum_{m=1}^{M} q_m^\top R_m(\theta)\right\}, \quad (6)$$

The above problem is a finite-sum convex-concave saddle-point problem if $R_m(\theta)$ is a convex function. However, introducing neural networks often renders $R_m(\theta)$ non-convex, leading to variance in algorithms like stochastic mirror descent and stochastic mirror prox, designed for finite-sum convex-concave saddle-point problems. This variance results in instability during the parameter learning process.

Here, inspired by (Carmon & Hausler, 2022; Yu et al., 2024), we incorporate variance reduction into the stochastic mirror prox algorithm. Specifically, we introduce a simple and effective task-sampling technique that selects $M$ samples, one for each task, to form the stochastic gradients. Additionally, we incorporate variance reduction strategies within each task to accelerate the convergence rate.

### 4.1. Distance-generating Function and Bregman Setups

Let $\|\cdot\|_x$ denote a general norm on a finite-dimensional Banach space $\mathcal{E}_x$, and its dual norm is defined as $\|\cdot\|_{x,*} = \sup_{y \in \mathcal{E}_x}\{\langle x, y \rangle \mid \|y\|_x \leq 1\}$. We define the set $[\mathcal{S}] = \{1, 2, \cdots, S\}$ and its index set $[\mathcal{S}]^0 = \{0, 1, \cdots, S-1\}$, where $S$ is a positive integer. For $(\theta, q) \in \Theta \times \mathcal{Q}_\alpha$ where $\mathcal{Q}_\alpha$ represents the simplex, we view $(\theta, q)$ as the concatenation of $\theta$ and $q$. The gradient $\nabla F_\alpha(\theta, q)$ is expressed as $\nabla F_\alpha(\theta, q) = (\nabla_\theta F_\alpha(\theta, q); -\nabla_q F_\alpha(\theta, q))$, denoting the combined gradient with respect to $\theta$ and $q$.

For primal-dual methods of the mirror descent type(Beck

& Teboulle, 2003), it is necessary to define a distance-generating function and the corresponding Bregman divergence.

**Definition 4.1.** (Distance-generating function) A continuous function $\psi_x : X \to \mathbb{R}$ is termed a distance-generating function with modulus $\alpha_x$ relative to the norm $\|\cdot\|_x$ if the following conditions hold: 1) The set $X^\circ = \{x \in X \mid \partial \psi_x(x) \neq 0\}$ is convex; 2) $\psi_x$ is continuously differentiable and $\alpha_x$-strongly convex with respect to $\|\cdot\|_x^2$, i.e., $\langle \nabla \psi_x(x_1) - \nabla \psi_x(x_2), x_1 - x_2 \rangle \geq \alpha_x \|x_1 - x_2\|_x^2, \quad \forall x_1, x_2 \in X^\circ$.

**Definition 4.2.** (Bregman divergence) The Bregman divergence $B_x : X \times X^\circ \to \mathbb{R}_+$ associated with the distance-generating function $\psi_x$ is defined as: $B_x(x, x^\circ) = \psi_x(x) - \psi_x(x^\circ) - \langle \nabla \psi_x(x^\circ), x - x^\circ \rangle$.

In this work, we equip $\Theta$ with a distance-generating function $\psi_\theta(\cdot)$, which has modulus $\alpha_\theta$ with respect to a norm $\|\cdot\|_\theta$ on $\mathcal{E}$. Similarly, $\psi_q(\cdot)$ is defined with modulus $\alpha_q$ relative to the norm $\|\cdot\|_q$. The choice of $\psi_x$ and $\|\cdot\|_x$ should align with the geometric structure of the domain. In this paper, we use $\psi_q(q) = \sum_{i=1}^{m} q_i \log q_i$, which corresponds to the entropy function.

The following standard assumptions are also widely used in Bregman setup analyses.

**Assumption 4.3.** (Boundedness of the Domain) The diameter of the domain $\Theta$ under the distance-generating function $\psi_\theta(\cdot)$ is bounded by a constant $D_\theta$, i.e., $\max_{\theta \in \Theta} \psi_\theta(\theta) - \min_{\theta \in \Theta} \psi_\theta(\theta) \leq D_\theta^2$. Similarly, the simplex $\mathcal{Q}_\alpha$ is assumed to be bounded by $D_q$. Since the entropy function $\psi_q$ is used, we have $D_q = \sqrt{\ln M}$.

We define the Cartesian product space $\mathcal{E} \times \mathbb{R}^M$ with its associated norm and dual norm as follows. For any $(\theta, q) \in \Theta \times \mathbb{R}^M$ and any $(\theta, q)^* = (\theta^*; q^*) \in \mathcal{E}^* \times \mathbb{R}^M$:

$$\|(\theta, q)\| := \sqrt{\frac{\alpha_\theta}{2D_\theta^2}\|\theta\|_\theta^2 + \frac{\alpha_q}{2D_q^2}\|q\|_q^2},$$

$$\|(\theta, q)^*\|_* := \sqrt{\frac{2D_\theta^2}{\alpha_\theta}\|\theta^*\|_{\theta,*}^2 + \frac{2D_q^2}{\alpha_q}\|q^*\|_{q,*}^2}. \tag{7}$$

The corresponding distance-generating function is defined as $\psi((\theta, q)) := \frac{1}{2D_\theta^2}\psi_\theta(\theta) + \frac{1}{2D_q^2}\psi_q(q)$.

It is straightforward to verify that $\psi((\theta, q))$ is 1-strongly convex with respect to the norm $\|\cdot\|$ defined in Eq. (7). Using this, we define the Bregman divergence $B : (\Theta \times \mathcal{Q}_\alpha) \times (\Theta \times \mathcal{Q}_\alpha)^\circ \to \mathbb{R}_+$ as: $B((\theta, q), (\theta, q)^\circ) := \psi((\theta, q)) - \psi((\theta, q)^\circ) - \langle \nabla \psi((\theta, q)^\circ), (\theta, q) - (\theta, q)^\circ \rangle$.

### 4.2. Double-loop Structure of Variance Reduction

Unfortunately, this method proves ineffective when stochastically optimizing CVaR due to the high variance in the

mini-batch gradient estimates of CVaR, particularly for non-convex problems like training deep neural networks. The double-loop structure of variance reduction is the commonly used strategy in optimization algorithms. It consists of an outer loop and an inner loop, which separate global updates from local refinements, providing a structured approach to achieving optimization goals. Inspired by the previous work (Carmon & Hausler, 2022; Alacaoglu & Malitsky, 2022), we adopt a double-loop structure based on variance reduction. The outer loop computes snapshot points in both the primal and dual spaces, while the inner loop uses a modified mirror prox method. Instead of computing two stochastic gradients, we use a combination of a mini-batch gradient and a stochastic gradient, building upon classical mirror prox algorithms.

**Outer Loop** The outer loop follows a standard variance reduction procedure (Johnson & Zhang, 2013), periodically computing snapshot points based on the weighted average of previous iterates. Let the snapshot in the $s$-th outer loop be $(\theta, q)^s$, and $\nabla \psi((\theta, q)^s)$ be the corresponding mirror snapshot. Formally, we define $(\theta, q)^*$ as :

$$(\theta, q)^s = \frac{\sum_{k=1}^{K_{s-1}} w_{k-1}^{s-1}(\theta, q)_k^{s-1}}{\sum_{k=1}^{K_s} w_{k-1}^{s-1}}, \tag{8}$$

and the mirror snapshot is

$$\nabla \psi\left((\theta, q)^s\right) = \frac{\sum_{k=1}^{K_{s-1}} w_{k-1}^{s-1} \nabla \psi\left((\theta, q)_k^{s-1}\right)}{\sum_{k=1}^{K_s} w_{k-1}^{s-1}}, \tag{9}$$

The full gradient $\nabla F_\alpha((\theta, q)^s)$ is then computed as:

$$\nabla_\theta F_\alpha(\theta^s, q^s) = \sum_{m=1}^{M} q_m^s \nabla R_m(\theta^s) \tag{10}$$

$$\nabla_q F_\alpha(\theta^s, q^s) = -\left[R_1(\theta^s), \cdots, R_M(\theta^s)\right]^\top$$

**Inner Loop** In the $k$-th inner loop, we updates $\mathbf{z}_k$ using a modified mirror prox step:

$$(\theta, q)_{k+1/2}^s = \arg \min_{(\theta, q) \in \mathcal{Z}} \Big\{ \langle \nabla F_\alpha((\theta, q)^s), (\theta, q) \rangle$$

$$+ w_k^s B((\theta, q), (\theta, q)^s) + (1 - w_k^s) B((\theta, q), (\theta, q)_k^s) \Big\}, \tag{11}$$

which utilizes the full gradient $\nabla F_\alpha(\theta^s, q^s)$. The above update employs techniques such as "negative momentum" (Driggs et al., 2022) to enhance convergence speed. Note that the above prox point using a full gradient differs from traditional mirror prox algorithms (Nemirovski, 2004), which use stochastic gradients. Although a full gradient can achieve more stable solutions, it is time-consuming. To alleviate computational complexity, we can replace the full gradient (Eq. (10)) with a mini-batch gradient.

## 4.3. Variance-reduced Stochastic Gradient Estimator

After $(\theta, q)^s_{k+1/2}$ is computed, we sample data points from each task (dataset) to construct the stochastic gradient. For the $m$-th task, we sample uniformly from the corresponding dataset $\mathcal{D}_{\tau,m}$. The sampling process is denoted as: 1) for each dataset, uniformly sampling one sample: $Z^s_{k,m} \sim U(\{Z_{m,i}\}^{n_m}_{i=1}), \forall m \in [M]$; 2) forming the task-specific samples as $Z^s_k := \{Z^s_{k,m}\}^M_{m=1}$.

The task sampling strategy ensures that stochastic gradients utilize information from all $M$ datasets. In this case, we can get the task-specific stochastic gradient based on the task-specific samples $\{Z^s_{k,m}\}^M_{m=1}$:

$$\nabla_\theta F_\alpha(Z^s_{k,m}; \theta^s, q^s) = \sum_{m=1}^M q^s_m \nabla \ell_m(Z^s_{k,m}; \theta^s)$$

$$\nabla_q F_\alpha(Z^s_{k,m}; \theta^s, q^s) = -\left[\ell_1(Z^s_{k,m}; \theta^s), \cdots, \ell_M(Z^s_{k,m}; \theta^s)\right]^\top$$

(12)

This formulation balances randomness across tasks, effectively leveraging the two-level finite-sum structure of Eq. (6).

Leveraging variance reduction techniques from (Johnson & Zhang, 2013), we construct a robust and efficient approach to minimize variance in stochastic gradients while maintaining computational efficiency. Specifically, we define the variance-reduced stochastic gradient estimator with the aid of the stochastic gradient at the $s$-th snapshot $(\theta, q)^s$ and the full gradient as follows:

$$g^s_k = \nabla F_\alpha\left(\{Z^s_{k,m}\}^M_{m=1}; (\theta, q)^s_{k+1/2}\right)$$
$$- \nabla F_\alpha\left(\{Z^s_{k,m}\}^M_{m=1}; (\theta, q)^s\right) + \nabla F_\alpha((\theta, q)^s).$$

(13)

where the second term $\nabla F_\alpha\left(\{Z^s_{k,m}\}^M_{m=1}; (\theta, q)^s\right)$ is given in Eq. (12), the third term $\nabla F_\alpha((\theta, q)^s)$ is given in Eq. (10). The above gradient is an unbiased estimator since the samples across all tasks are used.

With the aid of the mirror prox method, the next step is to compute $(\theta, q)^s_{k+1}$:

$$(\theta, q)^s_{k+1} = \arg\min_{(\theta, q) \in \Theta \times \mathcal{Q}_\alpha} \left\{\eta^s_k \langle g^s_k, (\theta, q)\rangle \right.$$
$$\left. + w^s_k B((\theta, q), (\theta, q)^s) + (1 - w^s_k) B((\theta, q), (\theta, q)^s_k)\right\}.$$

(14)

In the above stochastic gradient, the variance-reduced stochastic gradient estimator $g^s_k$ is used instead of the raw stochastic gradient at $(\theta, q)^s_k$ to achieve variance reduction. The final solution is given as:

$$(\theta, q)^S = \frac{\sum_{s=0}^{S-1} \sum_{k=0}^{K_s-1} \eta^s_k (\theta, q)^s_{k+1}}{\sum_{s=0}^{S-1} \sum_{k=0}^{K_s-1} \eta^s_k},$$

(15)

---

**Algorithm 1** Variance-Reduced Stochastic Mirror Prox Algorithm for Tail Task Risk Optimization

---

**Input**: Risk functions $\{\ell_m(\theta)\}_{m\in[M]}$ related to NPs, epoch number $S$, iteration numbers $\{K_s\}$, learning rates $\{\eta^s_k\}$, and weights $\{w^s_k\}$.

1: Initialize parameters $(\theta, q)_0 = (\theta_0, q_0) = \arg\min_{(\theta,q)\in\Theta\times\mathcal{Q}_\alpha} \psi((\theta, q))$ as the starting point.
2: **for** $s = 0$ to $S - 1$ **do**
3:     Compute the snapshot $(\theta, q)^s$ and the mirror snapshot $\nabla\psi((\theta, q)^s)$ according to Eq.(8) and Eq.(9), respectively.
4:     Compute the full gradient $\nabla F_\alpha((\theta, q)^s)$ according to Eq. (10). (mini-batch is feasible)
5:     **for** $k = 0$ to $K_s - 1$ **do**
6:         Compute $(\theta, q)^s_{k+1/2}$ according to Eq.(11).
7:         For each $m \in [M]$, $Z^s_{k,m} \sim U(\{Z_{m,i}\}^{n_m}_{i=1})$.
8:         Compute the variance-reduced stochastic gradient estimator $g^s_k$ defined in Eq.(13).
9:         Compute $(\theta, q)^s_{k+1}$ according to Eq.(14).
10:     **end for**
11:     Set $(\theta, q)^{s+1}_0 = (\theta, q)^s_{K_s}$.
12: **end for**
13: **Return** $(\theta, q)^S$ according to Eq. (15).

---

which adopts adaptive learning rates $\eta^s_k$ as weighting coefficients for the cumulated prox points. The whole variance-reduced stochastic mirror prox algorithm for tail task risk optimization is given in Algorithm 1.

### 4.4. Discussion

The task sampling method uniformly selects data across all tasks, thus reducing the variance of stochastic gradients. This approach leverages the hierarchical structure of the objective function to achieve a Lipschitz constant reduction, ultimately lowering the computational complexity. In the gradient computation process of the outer loop, the weighting coefficient for the gradient from the previous step is typically determined using an average weighting strategy, i.e., $w^{s-1}_{k-1} = \frac{1}{K_{s-1}}, \forall k = 1, \cdots, K_{s-1}$. However, we can set it as a learning rate $\eta^s_k$ as shown in Eq. (15). In many algorithms, the learning rate is gradually decreased, meaning that earlier iterations may contribute larger weights. This operation can reduce fluctuations in each gradient update and improve the stability of the algorithm. Additionally, by comprehensively leveraging historical solutions or gradients, the quality of the current solution can be enhanced. In summary, by using a variance-reduced gradient estimator in the inner loop, our solution significantly reduces gradient variance, improving convergence rates. The mirror prox technique ensures better handling of complex constraint sets and non-Euclidean geometries, making the optimization adaptable to various tasks. The final solution $(\theta, q)^S$ aggregates the

*Table 1.* Comparison of our robust NPs with the baselines on log-likelihood of the target points on two real-world datasets: CelebA and EMNIST. We train each method with 5 different seeds and report the mean and standard deviation.

| Method | CelebA | EMNIST | |
|---|---|---|---|
| | | Seen (0-9) | Unseen (10-46) |
| CNP | $2.1601_{\pm0.004}$ | $0.7373_{\pm0.004}$ | $0.4854_{\pm0.004}$ |
| BCNP | $2.1652_{\pm0.005}$ | $0.7552_{\pm0.005}$ | $0.4982_{\pm0.006}$ |
| SCNP | $2.1713_{\pm0.004}$ | $0.7631_{\pm0.006}$ | $0.5125_{\pm0.008}$ |
| RCNP | $\mathbf{2.1896}_{\pm0.005}$ | $\mathbf{0.7879}_{\pm0.005}$ | $\mathbf{0.5362}_{\pm0.006}$ |
| NP | $2.4816_{\pm0.015}$ | $0.7953_{\pm0.002}$ | $0.5847_{\pm0.003}$ |
| BNP | $2.7691_{\pm0.003}$ | $0.8706_{\pm0.005}$ | $\mathbf{0.7167}_{\pm0.012}$ |
| SNP | $2.8714_{\pm0.006}$ | $0.8873_{\pm0.003}$ | $0.7021_{\pm0.009}$ |
| RNP | $\mathbf{2.8915}_{\pm0.006}$ | $\mathbf{0.8911}_{\pm0.005}$ | $0.7121_{\pm0.006}$ |
| ANP | $2.9214_{\pm0.004}$ | $0.9815_{\pm0.006}$ | $0.8843_{\pm0.003}$ |
| BANP | $2.9411_{\pm0.008}$ | $0.9833_{\pm0.007}$ | $0.8891_{\pm0.003}$ |
| SANP | $2.9518_{\pm0.004}$ | $0.9753_{\pm0.008}$ | $0.8924_{\pm0.005}$ |
| RANP | $\mathbf{2.9718}_{\pm0.006}$ | $\mathbf{0.9881}_{\pm0.004}$ | $\mathbf{0.8967}_{\pm0.005}$ |
| ConvNP | $3.1241_{\pm0.004}$ | $1.1525_{\pm0.021}$ | $1.0351_{\pm0.006}$ |
| BConvNP | $3.1683_{\pm0.005}$ | $1.1651_{\pm0.009}$ | $1.0511_{\pm0.004}$ |
| SConvNP | $3.2135_{\pm0.006}$ | $1.2125_{\pm0.006}$ | $1.0698_{\pm0.008}$ |
| RConvNP | $\mathbf{3.2319}_{\pm0.005}$ | $\mathbf{1.2348}_{\pm0.005}$ | $\mathbf{1.0721}_{\pm0.006}$ |
| TNP | $4.4042_{\pm0.020}$ | $1.5501_{\pm0.004}$ | $1.4196_{\pm0.006}$ |
| BTNP | $4.4151_{\pm0.004}$ | $1.5561_{\pm0.003}$ | $1.4221_{\pm0.004}$ |
| STNP | $4.4124_{\pm0.003}$ | $1.5547_{\pm0.002}$ | $1.4236_{\pm0.003}$ |
| RTNP | $\mathbf{4.4226}_{\pm0.005}$ | $\mathbf{1.5572}_{\pm0.002}$ | $\mathbf{1.4413}_{\pm0.004}$ |

*Table 2.* Bayesian optimization experiments on data generated by different GP kernels

| Method | RBF | Matérn 5/2 | Periodic |
|---|---|---|---|
| ANP | $0.1245_{\pm0.003}$ | $0.1518_{\pm0.003}$ | $0.1892_{\pm0.002}$ |
| BANP | $0.1341_{\pm0.003}$ | $0.1316_{\pm0.004}$ | $0.1788_{\pm0.005}$ |
| SANP | $0.1142_{\pm0.002}$ | $0.1201_{\pm0.002}$ | $\mathbf{0.1672}_{\pm0.001}$ |
| RANP | $\mathbf{0.1025}_{\pm0.002}$ | $\mathbf{0.1171}_{\pm0.003}$ | $0.1779_{\pm0.006}$ |
| TNP | $0.1125_{\pm0.003}$ | $0.1451_{\pm0.001}$ | $0.1715_{\pm0.003}$ |
| BTNP | $0.1037_{\pm0.006}$ | $0.1455_{\pm0.003}$ | $0.1691_{\pm0.008}$ |
| STNP | $0.0998_{\pm0.004}$ | $0.1351_{\pm0.006}$ | $0.1561_{\pm0.002}$ |
| RTNP | $\mathbf{0.0915}_{\pm0.004}$ | $\mathbf{0.1289}_{\pm0.003}$ | $\mathbf{0.1463}_{\pm0.004}$ |

updates through weighted averaging, enabling a stable and high-quality output.

# 5. Experiments

We started with learning predictive functions on synthetic datasets, and high-dimensional tasks, e.g., image completion, Bayesian optimization, and contextual bandits, were performed to evaluate the properties of the NP-related models. We compare the baselines NP classes (CNP (Garnelo et al., 2018a), NP (Garnelo et al., 2018b), ANP (Kim et al., 2019), ConvNP (Foong et al., 2020), and TNP (Nguyen & Grover, 2022)), bootstrapping versions (Lee et al., 2020) (BCNP, BNP, BANP, BConvNP, and BTNP), stable versions (Liu et al., 2024a) (SCNP, SNP, SANP, SConvNP, and STNP) to our robust versions (RCNP, RNP, RANP, RConvNP, and RTNP).

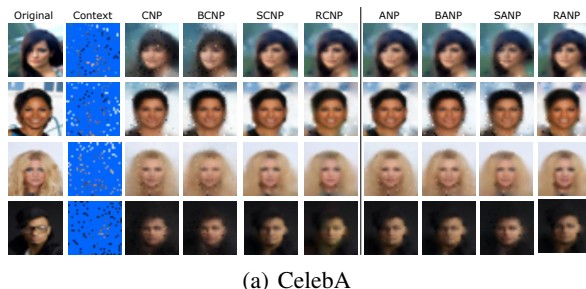

(a) CelebA

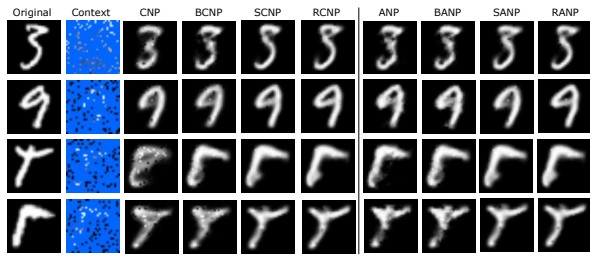

(b) EMNIST

*Figure 2.* Visualizations of CNP, ANP, and their bootstrapping versions (BCNP, BANP), stable versions (SNP, SANP), and robust versions (RCNP, RANP) for image completion tasks.

## 5.1. Image Completion

Following (Kim et al., 2019), we compared the models on image completion tasks on CelebA (Liu et al., 2015) and EMNIST (Cohen et al., 2017), where each image is downsampled to $32 \times 32$. More detailed experimental settings are given in the Appendix. Table 1 lists the log-likelihood of the target point of our robust NPs over the baselines on two datasets. Notably, on the EMNIST dataset, in addition to the standard experiments, we also conducted model-data mismatch experiments. Specifically, we trained on the first 10 classes and tested not only on these first 10 classes (seen classes) but also on the remaining 37 classes (unseen classes). Similar to 1D regression, our robust NPs still outperform all baselines, especially in model-data mismatch settings (unseen classes). In the mismatch setting, all the models become less accurate, but NPs with our robust solutions are affected less, which is similar to the results in 1D regression tasks. Furthermore, we visualize the generated images and Figure 2 shows that the proposed robust NPs produce noticeably better-completed images than the baselines.

## 5.2. Bayesian Optimization

Following the setting in BANP (Lee et al., 2020; Nguyen & Grover, 2022), we conducted the Bayesian optimization experiment, see Appendix for details. Taking GP data with RBF, Matérn 5/2, and Periodic prior functions as examples, we gave the results of ANP, TNP, corresponding bootstrapping NPs (BANP, BTNP), stable NPs (SANP, STNP), and

*Table 3.* Comparison of our robust NPs with the baselines on cumulative regret on contextual bandit problems with different values of $\gamma$. We run each model 50 times for each value of $\gamma$ and report the mean and standard deviation.

| Method | $\gamma = 0.7$ | $\gamma = 0.9$ | $\gamma = 0.95$ | $\gamma = 0.99$ | $\gamma = 0.995$ | $\gamma = 0.999$ |
|---|---|---|---|---|---|---|
| CNP | $4.0816_{\pm 0.321}$ | $8.1512_{\pm 0.411}$ | $8.0125_{\pm 0.393}$ | $26.771_{\pm 0.791}$ | $38.833_{\pm 0.986}$ | $93.2232_{\pm 2.564}$ |
| BCNP | $5.8921_{\pm 0.516}$ | $10.2726_{\pm 1.153}$ | $10.0116_{\pm 1.075}$ | $29.6628_{\pm 1.216}$ | $42.2516_{\pm 1.151}$ | $95.5624_{\pm 3.616}$ |
| SCNP | $3.5266_{\pm 0.415}$ | $8.29152_{\pm 0.525}$ | $\mathbf{7.2852}_{\pm 0.258}$ | $24.5264_{\pm 0.619}$ | $37.4265_{\pm 0.946}$ | $89.8256_{\pm 3.669}$ |
| RCNP | $\mathbf{3.1876}_{\pm 0.364}$ | $\mathbf{7.3115}_{\pm 0.657}$ | $7.6796_{\pm 0.647}$ | $\mathbf{23.4416}_{\pm 0.547}$ | $\mathbf{35.5597}_{\pm 0.863}$ | $\mathbf{87.1514}_{\pm 2.165}$ |
| NP | $1.5626_{\pm 0.125}$ | $2.9616_{\pm 0.291}$ | $4.2513_{\pm 0.251}$ | $18.2417_{\pm 0.465}$ | $25.5646_{\pm 0.195}$ | $62.7342_{\pm 1.503}$ |
| BNP | $62.5115_{\pm 1.076}$ | $57.4966_{\pm 2.138}$ | $58.2276_{\pm 2.285}$ | $58.9122_{\pm 3.785}$ | $62.5021_{\pm 5.156}$ | $77.5255_{\pm 6.226}$ |
| SNP | $\mathbf{1.4356}_{\pm 0.214}$ | $\mathbf{2.4451}_{\pm 0.552}$ | $3.4056_{\pm 0.242}$ | $17.2551_{\pm 0.357}$ | $23.5226_{\pm 0.226}$ | $60.8862_{\pm 0.952}$ |
| RNP | $1.4581_{\pm 0.153}$ | $2.5632_{\pm 0.366}$ | $\mathbf{3.2367}_{\pm 0.195}$ | $\mathbf{15.6269}_{\pm 1.329}$ | $\mathbf{22.5154}_{\pm 0.851}$ | $\mathbf{58.2676}_{\pm 1.124}$ |
| ANP | $1.6246_{\pm 0.172}$ | $4.0558_{\pm 0.316}$ | $5.3983_{\pm 0.506}$ | $19.5712_{\pm 0.678}$ | $27.6516_{\pm 0.952}$ | $73.3661_{\pm 5.956}$ |
| BANP | $5.3165_{\pm 15.614}$ | $15.2415_{\pm 19.156}$ | $34.1167_{\pm 24.167}$ | $58.2314_{\pm 23.662}$ | $62.3364_{\pm 17.994}$ | $69.3436_{\pm 18.362}$ |
| SANP | $1.5878_{\pm 0.205}$ | $3.6691_{\pm 0.564}$ | $4.8767_{\pm 0.704}$ | $18.2519_{\pm 0.552}$ | $25.1169_{\pm 0.891}$ | $71.5169_{\pm 3.537}$ |
| RANP | $\mathbf{1.2555}_{\pm 0.162}$ | $\mathbf{2.6256}_{\pm 0.536}$ | $\mathbf{3.8987}_{\pm 0.663}$ | $16.5155_{\pm 0.641}$ | $24.1519_{\pm 0.983}$ | $70.1414_{\pm 1.368}$ |
| ConvNP | $1.5035_{\pm 0.104}$ | $2.4552_{\pm 0.317}$ | $3.1761_{\pm 0.447}$ | $16.3585_{\pm 0.988}$ | $25.2359_{\pm 1.251}$ | $72.3526_{\pm 4.257}$ |
| BConvNP | $2.2563_{\pm 0.251}$ | $3.6762_{\pm 0.376}$ | $4.4626_{\pm 0.551}$ | $18.2516_{\pm 1.121}$ | $26.3629_{\pm 1.357}$ | $74.6324_{\pm 4.257}$ |
| SConvNP | $1.3846_{\pm 0.536}$ | $2.0081_{\pm 0.208}$ | $2.2662_{\pm 0.226}$ | $15.2315_{\pm 0.796}$ | $23.5266_{\pm 1.447}$ | $70.1251_{\pm 3.654}$ |
| RConvNP | $\mathbf{1.1536}_{\pm 0.103}$ | $\mathbf{1.7369}_{\pm 0.249}$ | $\mathbf{1.9854}_{\pm 0.341}$ | $\mathbf{13.2352}_{\pm 0.627}$ | $\mathbf{19.2516}_{\pm 2.352}$ | $\mathbf{61.1273}_{\pm 2.226}$ |
| TNP | $1.1895_{\pm 0.942}$ | $1.7057_{\pm 0.418}$ | $2.5562_{\pm 0.438}$ | $3.5795_{\pm 1.225}$ | $4.6867_{\pm 1.092}$ | $9.5694_{\pm 0.465}$ |
| BTNP | $3.3686_{\pm 0.356}$ | $2.7356_{\pm 0.523}$ | $4.6371_{\pm 0.628}$ | $5.2526_{\pm 1.643}$ | $7.3521_{\pm 1.367}$ | $11.2657_{\pm 2.277}$ |
| STNP | $\mathbf{1.1637}_{\pm 0.128}$ | $\mathbf{1.4362}_{\pm 0.378}$ | $2.1526_{\pm 0.574}$ | $2.3612_{\pm 1.522}$ | $3.5267_{\pm 1.267}$ | $8.9824_{\pm 0.678}$ |
| RTNP | $1.2155_{\pm 0.204}$ | $1.5962_{\pm 0.539}$ | $\mathbf{1.1371}_{\pm 0.348}$ | $\mathbf{2.0813}_{\pm 0.896}$ | $\mathbf{3.1231}_{\pm 0.957}$ | $\mathbf{7.2355}_{\pm 1.429}$ |

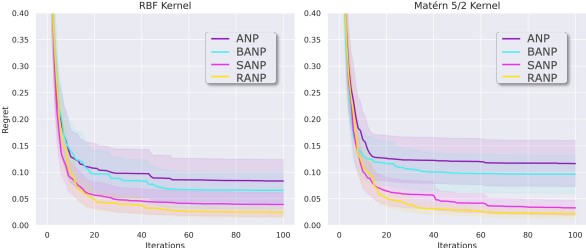

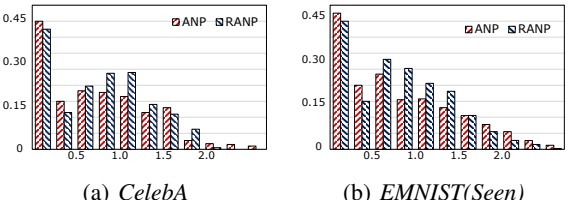

(a) *CelebA*  (b) *EMNIST(Seen)*

*Figure 4.* The histograms of task risks in image completion tasks.

*Figure 3.* Regret performance on 1D Bayesian optimization tasks. For each kernel, we generate 100 functions and report the mean and standard deviation.

robust NPs (RANP, RTNP). Note that we use NPs as the surrogate functions and Upper Confidence Bound (UCB) as the acquisition function. To maintain consistent comparison, we standardized the initializations and normalized the results. For each objective function, we run Bayesian optimization for 100 iterations, and simple regret is used as the evaluation metric. As shown in Table 2 and Figure 3, we can see that our robust solutions consistently achieve lower regret than other NPs.

## 5.3. Contextual Bandits

Following (Riquelme et al., 2018; Nguyen & Grover, 2022), we focus on the contextual bandit problem, which is a type of reinforcement learning problem where an agent makes a series of decisions, each with an associated context, to maximize cumulative reward. More detailed experimental settings, including problem definition, training, and testing procedure can be given in the Appendix. Considering this

problem involves dividing a fixed-sized circle into a low-reward region (the central circle) and four high-reward regions (the remaining outer ring divided into four equal parts), a control variable $\gamma$ is introduced to manage the proportion between the high and low-reward regions. The higher the $\gamma$ value, the larger the central circle and the smaller the outer ring, and vice versa. We conducted experiments with different $\gamma$, and Table 3 presents the experimental results of all methods. We can see that NPs with our robust solutions outperform all baselines by a large margin in most settings, especially for harder settings (larger value of $\gamma$). We obtained similar experimental results as in (Nguyen & Grover, 2022), where the performance of models using the attention mechanism is generally lower than that of models without the attention mechanism, which is contrary to the results obtained in other experimental tasks.

## 5.4. Risk Distribution

As we stated previously, the proposed strategy can iteratively reshape the task risk distribution to increase robustness, i.e., transport the probability mass in high-risk regions

to the lower-risk regions. To verify this point, we present the task risk distributions with and without using our strategy. Specifically, taking RANP as an example, Figure 4 shows the task risk distributions of ANP and RANP on image completion tasks with two different kernels. We select MSE as the risk measure. Overall, it can be seen that the robust solutions shift the original risk distribution to the left, reducing the proportion of high-risk tasks and effectively demonstrating the robustness of the proposed model.

## 6. Conclusion and Future Work

The paper introduces a novel approach to enhance the robustness of Neural Processes (NPs) in skill transfer learning across different tasks. Traditional NPs focus on empirical risk minimization, which can lead to varying levels of fast adaptation, potentially resulting in catastrophic outcomes in risk-sensitive tasks. To address this problem, we propose incorporating controllable expected tail meta risk, allowing control over the worst-case fast adaptation scenarios at a given probabilistic level. By reformulating the risk minimization as a distributional robust formulation, we introduce a variance-reduced stochastic mirror prox algorithm with double-loop structures and leverage the two-level structure to get a stochastic gradient among samples across all tasks. The proposed risk and optimization strategy is versatile, working with various NPs while providing a less biased approximation.

A common optimization approach for the minimax problem of CVaR involves leveraging a two-player game framework. Designing efficient methods to optimize the game between the two players is a topic worth exploring. Although our variance-reduced stochastic mirror prox algorithm has been validated for its effectiveness both intuitively and experimentally, this kind of algorithm is primarily designed for convex optimization. Models with deep neural networks typically involve high-dimensional, non-convex objectives for which the method lacks general convergence guarantees. One future point is to investigate the algorithms' convergence guarantee. Additionally, improving the efficiency of the overall optimization process is another point of interest.

## Acknowledgements

This work was partly supported by the Talent Fund of Beijing Jiaotong University (2024XKRC075); Beijing Natural Science Foundation (4244096); The National Natural Science Foundation of China under Grant (62406019, 62436001, 62176020); The Joint Foundation of the Ministry of Education for Innovation team (8091B042235); The National Key Research and Development Program of China (2024YFE0202900); The State Key Laboratory of Rail Traffic Control and Safety (RCS2023K006); Natural Science Foundation of Chongqing (Grant CSTB2023NSCQ-MSX1020).

## Impact Statement

This paper presents work whose goal is to advance the field of Machine Learning. There are many potential societal consequences of our work, none which we feel must be specifically highlighted here.

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

# A. Basic Knowledge

## A.1. CVaR

we recall standard definitions of Value-at-Risk and Conditional Value-at-Risk which are used in our solution.

**Definition A.1.** ((Conditional) Value-at-Risk) Given a random variable $X$ with cumulative distribution function $F(\cdot)$, the Value-at-Risk VaR $v_\alpha(X)$ and Conditional Value-at-Risk CVaR $c_\alpha(X)$ at level $\alpha \in (0,1)$ are defined as:

$$
\begin{aligned}
v_\alpha(X) &= \inf\{\xi : p(X \leq \xi) \geq \alpha\}, \text{ and} \\
c_\alpha(X) &= v_\alpha(X) + \frac{1}{1-\alpha}\mathbb{E}[X - v_\alpha(X)]^+,
\end{aligned}
\tag{16}
$$

where the notation $[\cdot]^+ = \max(0, \cdot)$ and $\alpha$ is a predefined confidence level or risk level.

In machine learning, the concept of Conditional Value at Risk (CVaR) has been widely applied. For instance, the $\nu$-SVM algorithm (Schölkopf et al., 2000), can be seen as addressing the minimization of the CVaR of the loss, as analyzed by (Gotoh & Takeda, 2016). Additionally, (Shalev-Shwartz & Wexler, 2016) suggests focusing on minimizing the maximum loss across all samples, which is equivalent to the limiting case of CVaR when $\alpha \to 0$. Building on this, (Fan et al., 2017) extend the idea to optimize a top-$k$ average loss. Although they do not explicitly connect their criterion to CVaR, their learning strategy aligns with the CVaR framework for empirical measures. For optimization tasks, (Ogryczak & Tamir, 2003) proposed a method to minimize the maximum value of a summation over $k$ functions. This approach is closely related to the "truncated" algorithm developed by (Rockafellar et al., 2000) for CVaR optimization. CVaR has since found applications in areas such as risk-averse reinforcement learning, including works by (Sani et al., 2012) and (Chow et al., 2018), and fairness in machine learning, as discussed by (Williamson & Menon, 2019). Despite its widespread use, the original CVaR optimization approach suffers from challenges such as the high variance of mini-batch gradient estimators (Rockafellar et al., 2000). To overcome these limitations, adaptive sampling techniques (Shalev-Shwartz & Wexler, 2016) have been proposed, enabling the efficient handling of large-scale datasets and complex models, including deep neural networks (Curi et al., 2020; Wang et al., 2024).

## A.2. Neural Processes

The neural process aims to learn a stochastic process (random function) mapping target features $\mathbf{x}_i$ to prediction $y_i$ given the context set $\mathcal{D}^{\mathcal{C}}$ as training data (a realization from the stochastic process), i.e., learning

$$
\log p\left(\mathbf{y}^{\mathcal{T}}|\mathbf{X}^{\mathcal{T}}, \mathcal{D}^{\mathcal{C}}\right) = \sum_{i=1}^{N} p\left(y_i|\mathbf{x}_i, \mathcal{D}^{\mathcal{C}}\right).
\tag{17}
$$

Conditional neural process (CNP) (Garnelo et al., 2018a) describes $p\left(y_i|\mathbf{x}_i, \mathcal{D}^{\mathcal{C}}\right)$ with a deterministic neural network taking $\mathcal{D}^{\mathcal{C}}$ to output the parameters of $p\left(y_i|\mathbf{x}_i, \mathcal{D}^{\mathcal{C}}\right)$. CNP consists of an encoder $f_{\text{enc}}(\cdot)$, an aggregator $f_{\text{agg}}(\cdot)$ and a decoder $f_{\text{dec}}(\cdot)$; the encoder summarizes $\mathcal{D}^{\mathcal{C}}$ and $\mathbf{x}_i$ into latent representations $[\mathbf{r}_1, \cdots, \mathbf{r}_{|\mathcal{C}|}] \in \mathbb{R}^{|\mathcal{C}| \times d}$ via permutation-invariant neural network (Zaheer et al., 2017), where $d$ is the number of latent dimensions, and aggregator summarizes the encoded context features to a single representation $\mathbf{r}^{\mathcal{C}}$, and decoder takes as input the aggregated representations $\mathbf{r}^{\mathcal{C}}$ and $\mathbf{x}_i$ and output the single output-specific mean $\mu_i$ and variance $\sigma_i^2$ for the corresponding value of $y_i$.

$$
\begin{aligned}
\mathbf{r}_i &= f_{\text{enc}}\left(\mathbf{x}_i, y_i\right), \quad i \in \mathcal{C} \\
\mathbf{r}^{\mathcal{C}} &= \frac{1}{|\mathcal{C}|}\sum_{i \in \mathcal{C}} \mathbf{r}_i \\
\phi &= f_{\text{agg}}(\mathbf{r}^{\mathcal{C}}) \\
(\mu_i, \sigma_i) &= f_{\text{dec}}(\phi, \mathbf{x}_i), \\
p\left(y_i|\mathbf{x}_i, \mathcal{D}^{\mathcal{C}}\right) &= \mathcal{N}\left(y_i; \mu_i, \sigma_i^2\right) \quad i \in \mathcal{T}
\end{aligned}
\tag{18}
$$

where $f_{\text{enc}}(\cdot)$ and $f_{\text{dec}}(\cdot)$ are feed-forward neural networks. The decoder output $\mu_i$ and variance $\sigma_i^2$ are predicted mean and variance. We use *Gaussian* distribution $\mathcal{N}(y_i; \mu_i, \sigma_i^2)$ as predictive distribution. CNP is trained to maximize the expected likelihood $\mathbb{E}_{p(\mathcal{T})}[p\left(y_i|\mathbf{x}_i, \mathcal{D}^{\mathcal{C}}\right)]$.

Neural process (Garnelo et al., 2018b) further models functional uncertainty using a global latent variable. Unlike CNP, which maps a context into a deterministic representation $\widetilde{\mathbf{r}}_i$, NP encoders a context into a *Gaussian* latent variable $\mathbf{z}$, giving additional stochasticity in function construction. Following (Kim et al., 2019), we consider an NP with both a deterministic path and latent path, where the deterministic path models the overall skeleton of the function $\widetilde{\mathbf{r}}_i$, and the latent path models the functional uncertainty:

$$
\begin{aligned}
\mathbf{r}_i &= f_{\text{enc}}^{(1)}\left(\mathbf{x}_i, y_i\right), \quad i \in \mathcal{C} \\
\mathbf{r}^{\mathcal{C}} &= \frac{1}{|\mathcal{C}|} \sum_{i \in \mathcal{C}} \mathbf{r}_i \\
\phi &= f_{\text{agg}}(\mathbf{r}) \\
(\mu_z, \sigma_z) &= f_{\text{enc}}^{(2)}\left(\mathcal{D}^{\mathcal{C}}\right), \\
q(\mathbf{z}|\mathcal{D}^{\mathcal{C}}) &= \mathcal{N}(\mathbf{z}; \mu_z, \sigma_z^2) \\
(\mu_i, \sigma_i) &= f_{\text{dec}}(\phi, \mathbf{z}, \mathbf{x}_i), \\
p\left(y_i|\mathbf{x}_i, \mathbf{z}, \mathcal{D}^{\mathcal{C}}\right) &= \mathcal{N}\left(y_i; \mu_i, \sigma_i^2\right) \quad i \in \mathcal{T}
\end{aligned}
\tag{19}
$$

with $f_{\text{enc}}^{(1)}(\cdot)$ and $f_{\text{enc}}^{(2)}(\cdot)$ having the same structure as $f_{\text{enc}}(\cdot)$ in Eq.(18). In this scenario, the conditional distribution is lower bounded as:

$$
\log p\left(\mathbf{y}|\mathbf{X}, \mathcal{D}^{\mathcal{C}}\right) \geq \sum_{i=1}^{N} \mathbb{E}_{q(\mathbf{z}|\mathcal{D}^{\mathcal{C}})}\left[\log \frac{p\left(y_i|\mathbf{x}_i, \mathbf{z}, \mathcal{D}^{\mathcal{C}}\right) P(\mathbf{z}|\mathcal{D}^{\mathcal{C}})}{q(\mathbf{z}|\mathbf{X}, \mathbf{y})}\right].
\tag{20}
$$

We further approximate $q(\mathbf{z}|\mathcal{D}^{\mathcal{C}}) \approx p(\mathbf{z}|\mathcal{D}^{\mathcal{C}})$ and train the model by maximizing this expected lower bound over tasks. Furthermore, ANP introduces attention mechanisms into NP to resolve the issue of under-fitting.

## B. Theoretical Investigations

**Proposition B.1.** *(Curi et al., 2020) Let $h : \mathcal{X} \to \mathcal{Y}$ be a finite function class $|\mathcal{H}|$. Let $\ell(h) : \mathcal{H} \to [0, 1]$ be a random variable. Then, for any $0 < \alpha \leq 1$, with probability at least $1 - \delta$,*

$$
\mathbb{E}\left[\sup_{h \in \mathcal{H}} \left|\hat{R}_{\alpha}(s, \theta) - R_{\alpha}(s, \theta)\right|\right] \leq \frac{1}{1-\alpha} \sqrt{\frac{\log(2|\mathcal{H}|/\delta)}{MN}}.
\tag{21}
$$

To deeply analyze the Bregman setup, we give the following assumptions.

**Assumption B.2.** (Boundedness of the Domain) The diameter of the domain $\Theta$ under the distance-generating function $\psi_{\theta}(\cdot)$ is bounded by a constant $D_{\theta}$, i.e.,

$$
\max_{\theta \in \Theta} \psi_{\theta}(\theta) - \min_{\theta \in \Theta} \psi_{\theta}(\theta) \leq D_{\theta}^2.
\tag{22}
$$

Similarly, the simplex $\mathcal{Q}_{\alpha}$ is assumed to be bounded by $D_q$. Since the entropy function $\psi_q$ is used, we have $D_q = \sqrt{\ln M}$.

**Assumption B.3.** ((Smoothness and Lipschitz Continuity)) For any sample from meta-dataset (dataset of datasets) $\mathcal{D}_{\tau, 1:M} = \{\mathcal{D}_{\tau, m}\}_{m=1}^{M}$, the loss function $\ell_m(\theta)$ is assumed to satisfy $L$-smoothness and $G$-Lipschitz continuity.

For the task-specific stochastic gradient defined in Eq. (12), we have the following theorem to indicate its unbiasedness.

**Theorem B.4.** *The stochastic gradient*

$$
\nabla F_{\alpha}(Z_{k,m}^s; \theta^s, q^s) = \begin{bmatrix} \nabla_{\theta} F_{\alpha}(Z_{k,m}^s; \theta^s, q^s) \\ \nabla_q F_{\alpha}(Z_{k,m}^s; \theta^s, q^s) \end{bmatrix} = \begin{bmatrix} \sum_{m=1}^{M} q_m^s \nabla \ell_m(Z_{k,m}^s; \theta^s) \\ -\left[\ell_1(Z_{k,m}^s; \theta^s), , \cdots, \ell_M(Z_{k,m}^s; \theta^s)\right]^{\top} \end{bmatrix}
\tag{23}
$$

*defined in Eq. (12) is unbiased.*

*Proof.* For the sampling strategy defined in Section 4.3: 1) for each dataset, uniformly sampling one sample: $Z_{k,m}^s \sim U(\{Z_{m,i}\}_{i=1}^{n_m}), \forall m \in [M]$; 2) forming the task-specific samples as $\{Z_{k,m}^s\}_{m=1}^{M}$, we have

$$
\forall(\theta, q) \in \Theta \times \mathcal{Q}_{\alpha}: \quad \mathbb{E}[\ell_m(Z_{k,m}^s; \theta)] = R_m(\theta), \quad \mathbb{E}[\nabla \ell_m(Z_{k,m}^s; \theta)] = \nabla R_m(\theta).
\tag{24}
$$

Based on the additivity of expectation, we can get

$$\forall (\theta, q) \in \Theta \times \mathcal{Q}_\alpha : \quad \mathbb{E}[\nabla F_\alpha(Z_k^s; \theta)] = \nabla F_\alpha(\theta). \tag{25}$$

In this case, it can be proved that the stochastic gradient is unbiased. $\qquad\square$

We assume that $\ell_m(Z_{k,m}; \theta)$ is $L$-smooth and $G$-Lipschitz continmuous for all $m \in [M], k \in [n_m]$. The following theorem analyzes the Lipschitz continuity of the stochastic gradient $\nabla F_\alpha(Z_{k,m}^s; \theta^s, q^s)$.

**Theorem B.5.** *For any $s \in [S]^0$, $k \in [K_s]^0$, $\nabla F_\alpha(Z_{k,m}^s; \theta^s, q^s)$ is $L^*$-Lipschitz continuous, where $L_* = 2D_\theta \max\{\sqrt{2D_\theta^2 L^2 + G^2 \ln M}, G\sqrt{2\ln M}\}$.*

*Proof.* Randomly selecting two parameters $(\theta^+, q^+)$ and $(\theta^-, q^-)$, we can bound the gradient of $\theta$ as follows:

$$\begin{aligned}
&\|\nabla_\theta F_\alpha(Z_k^s; \theta^+, q^+) - \nabla_\theta F_\alpha(Z_k^s; \theta^-, q^-)\|_{\theta a, *}^2 \\
=& \left\| \sum_{m=1}^M q_m^+ [\nabla \ell_m(Z_{k,m}^s; \theta^+) - \nabla \ell_m(Z_{k,m}^s; \theta^-)] + \sum_{m=1}^M (q_m^+ - q_m^-) \nabla \ell_m(Z_{k,m}^s; \theta^-) \right\|_{\theta, *}^2 \\
\leq& 2 \left\| \sum_{m=1}^M q_m^+ [\nabla \ell_m(Z_{k,m}^s; \theta^+) - \nabla \ell_m(Z_{k,m}^s; \theta^-)] \right\|_{\theta, *}^2 + 2 \left\| \sum_{m=1}^M (q_m^+ - q_m^-) \nabla \ell_m(Z_{k,m}^s; \theta^-) \right\|_{\theta, *}^2 \\
\leq& 2 \sum_{m=1}^M q_m^+ \left\| \nabla \ell_m(Z_{k,m}^s; \theta^+) - \nabla \ell_m(Z_{k,m}^s; \theta^-) \right\|_{\theta, *}^2 + 2 \left( \sum_{m=1}^M |q_m^+ - q_m^-| \left\| \nabla \ell_m(Z_{k,m}^s; \theta^-) \right\|_{\theta, *}^2 \right) \\
\leq& 2 \sum_{m=1}^M q_m^+ L^2 \|\theta^+ - \theta^-\|_\theta^2 + 2 \left( \sum_{m=1}^M |q_m^+ - q_m^-| G \right) \\
=& 2L^2 \|\theta^+ - \theta^-\|_\theta^2 + 2G^2 \|q^+ - q^-\|_1^2.
\end{aligned} \tag{26}$$

We can also bound the gradient of $q$ as follows:

$$\forall m \in [M] : \quad \ell_m(Z_{k,m}^s; \theta^+) - \ell_m(Z_{k,m}^s; \theta^-) \leq G \|\theta^+ - \theta^-\|_\theta. \tag{27}$$

Given $\nabla_q F_\alpha(Z_{k,m}^s; \theta^s, q^s) = -\left[\ell_1(Z_{k,m}^s; \theta^s), , \cdots, \ell_M(Z_{k,m}^s; \theta^s)\right]^\top$, we have

$$\begin{aligned}
&\|\nabla_q F_\alpha(Z_{k,m}^s; \theta^+, q^+) - \nabla_q F_\alpha(Z_{k,m}^s; \theta^-, q^-)\|_\infty^2 \\
=& \max_{m \in [M]} [\ell_m(Z_{k,m}^s; \theta^+) - \ell_m(Z_{k,m}^s; \theta^-)]^2 \\
\leq& G^2 \|\theta^+ - \theta^-\|_\theta.
\end{aligned} \tag{28}$$

Based on the above bound, we have

$$\begin{aligned}
&\|\nabla F_\alpha(Z_{k,m}^s; \theta^+, q^+) - \nabla F_\alpha(Z_{k,m}^s; \theta^-, q^-)\|_*^2 \\
=& 2D_\theta^2 \|\nabla_\theta F_\alpha(Z_{k,m}^s; \theta^+, q^+) - \nabla_\theta F_\alpha(Z_{k,m}^s; \theta^-, q^-)\|_{\theta, *}^2 + 2\ln M \|\nabla_q F_\alpha(Z_{k,m}^s; \theta^+, q^+) - \nabla_q F_\alpha(Z_{k,m}^s; \theta^-, q^-)\|_\infty^2 \\
\leq& (4D_\theta^2 L^2 + 2G^2 \ln M)\|\theta^+ - \theta^-\|_\theta^2 + 4D_\theta^2 G^2 \|q^+ - q^-\|_1^2 \\
\leq& \frac{1}{2D_\theta^2} \left[ (4D_\theta^2 L^2 + G^2 \ln M)\|\theta^+ - \theta^-\|_\theta^2 \right] + \frac{1}{2\ln M}[4D_\theta^2(2G^2 \ln M)\|q^+ - q^-\|_1^2] \\
\leq& \frac{L_*^2}{2D_\theta^2} \|\theta^+ - \theta^-\|_\theta^2 + \frac{L_*^2}{2\ln M} \|q^+ - q^-\|_1^2 \\
=& L_*^2 \|(\theta^+, q^+) - (\theta^-, q^-)\|^2
\end{aligned}$$

$$\tag{29}$$

$\qquad\square$

# C. Model Architecture

The architectural details of the CNP, NP, and ANP are the same as in (Kim et al., 2019). Here we give the detailed architectures of the encoder and decoder of NPs.

## C.1. Encoder without attention

Encoder focuses on learning embeddings for each data point in the context set, and the basic component is multi-layer perceptron, which is defined by

$$\text{MLP}(l, d_{in}, d_h, d_{out}) = \text{LINEAR}(d_h, d_{out}) \circ \underbrace{(\text{RELU} \circ \text{LINEAR}(d_h, d_h) \circ \cdots)}_{\times (l-2)} \circ \text{LINEAR}(d_h, d_{in}) \tag{30}$$

where $l$ is the number of layers, $d_{in}$, $d_h$ and $d_{out}$ are dimensinalities of inputs, hidden unites and outputs. Here $\text{RELU}(\cdot)$ is adapted as activation function.

The encoder in Vanilla CNP uses a deterministic encoder that focuses on learning embeddings for each data point in the context set.

$$\mathbf{r}_i = \text{MLP}(l_{e1}, d_x + d_y, d_h, d_h)([\mathbf{x}_i, y_i]),$$
$$\mathbf{r}^{\mathcal{C}} = \sum_{i \in \mathcal{C}} \mathbf{r}_i, \quad \phi = \text{MLP}(l_{e2}, d_h, d_h)(\mathbf{r}^{\mathcal{C}}) \tag{31}$$

where $d_x$ and $d_y$ are the dimensionalities of $\mathbf{x}_i$ and $y_i$.

To follow the encoder structure in NP, we introduce another encoder aligned with the original deterministic encoder to permit the same number of parameters, i.e.,

$$\mathbf{r}_i^{(1)} = \text{MLP}(l_{e1}, d_x + d_y, d_h, d_h)([\mathbf{x}_i, y_i])$$
$$\mathbf{r}_{\mathcal{C}}^{(1)} = \sum_{i \in \mathcal{C}} \mathbf{r}_i^{(1)}, \quad \phi_1 = \text{MLP}(l_{e2}, d_h, d_h)(\mathbf{r}_{\mathcal{C}}^{(1)})$$
$$\mathbf{r}_i^{(2)} = \text{MLP}(l_{e1}, d_x + d_y, d_h, d_h)([\mathbf{x}_i, y_i]) \tag{32}$$
$$\mathbf{r}_{\mathcal{C}}^{(2)} = \sum_{i \in \mathcal{C}} \mathbf{r}_i^{(2)}, \quad \phi_2 = \text{MLP}(l_{e2}, d_h, d_h)(\mathbf{r}_{\mathcal{C}}^{(2)})$$
$$\phi = [\phi_1, \phi_2]$$

The encoder in NP contains a deterministic path and a latent path, i.e.,

$$\mathbf{r}_i^{(1)} = \text{MLP}(l_{de1}, d_x + d_y, d_h, d_h)([\mathbf{x}_i, y_i])$$
$$\mathbf{r}_{\mathcal{C}}^{(1)} = \sum_{i \in \mathcal{C}} \mathbf{r}_i^{(1)}, \quad \phi = \text{MLP}(l_{de2}, d_h, d_h)(\mathbf{r}_{\mathcal{C}}^{(1)})$$
$$\mathbf{r}_i^{(2)} = \text{MLP}(l_{la1}, d_x + d_y, d_h, d_h)([\mathbf{x}_i, y_i]) \tag{33}$$
$$\mathbf{r}_{\mathcal{C}}^{(2)} = \sum_{i \in \mathcal{C}} \mathbf{r}_i^{(2)}, \quad [\mu_z, \sigma_z'] = \text{MLP}(l_{la2}, d_h, d_h)(\mathbf{r}_{\mathcal{C}}^{(2)})$$
$$\sigma_z = 0.1 + 0.9 \cdot \text{SIGMOID}(\sigma_z'), \quad \mathbf{z} \sim \mathcal{N}(\mu_z, \text{diag}(\sigma_z^2)).$$

In this case, the encoder outputs deterministic representation $\phi$ and latent representation $\mathbf{z}$.

## C.2. Encoder with attention

The attention mechanism is widely used in NPs, Specifically, multi-head attention (Vaswani et al., 2017) is adapted, which is defined by

$$
\begin{aligned}
\mathbf{Q}' &= \{\text{LINEAR}(d_q, d_{out})(\mathbf{q})\}_{\mathbf{q} \in \mathbf{Q}}, \\
\{\mathbf{Q}'_i\}_{i=1}^{n_{head}} &= \text{SPLIT}(\mathbf{Q}', n_{head}), \\
\mathbf{K}' &= \{\text{LINEAR}(d_k, d_{out})(\mathbf{k})\}_{\mathbf{k} \in \mathbf{K}}, \\
\{\mathbf{K}'_i\}_{i=1}^{n_{head}} &= \text{SPLIT}(\mathbf{K}', n_{head}), \\
\mathbf{V}' &= \{\text{LINEAR}(d_v, d_{out})(\mathbf{v})\}_{\mathbf{v} \in \mathbf{V}}, \\
\{\mathbf{V}'_i\}_{i=1}^{n_{head}} &= \text{SPLIT}(\mathbf{V}', n_{head}), \\
\mathbf{H}_i &= \text{SOFTMAX}\left(\mathbf{Q}'_i(\mathbf{K}'_i)^\top / \sqrt{d_{out}}\right)\mathbf{V}'_i, \\
\mathbf{H} &= \text{CONCAT}\left(\{\mathbf{H}_i\}_{i=1}^{n_{head}}\right) \\
\mathbf{H}' &= \text{LAYERNORM}(\mathbf{Q}' + \mathbf{H}) \\
\text{MHA}(d_{out})(\mathbf{Q}, \mathbf{K}, \mathbf{V}) &= \text{LAYERNORM}(\mathbf{H}' + \text{RELU}(\text{LINEAR}(d_{out}, d_{out})))
\end{aligned}
\tag{34}
$$

where $d_q, d_v, d_k$ are the dimensionalities of query $\mathbf{Q}$, key $\mathbf{K}$, and value $\mathbf{V}$, respectively. $n_{head}$ is the number of head. Here Layer normalization (Ba et al., 2016) LAYERNORM($\cdot$) is adapted. It is easy to derive self-attention by setting $\mathbf{Q} = \mathbf{K} = \mathbf{V}$, i.e.,

$$
\text{SA}(d_{out}))(\mathbf{X}) = \text{MHA}(d_{out})(\mathbf{X}, \mathbf{X}, \mathbf{X})
\tag{35}
$$

For CNP, the encoder with attention still contains two deterministic paths,

$$
\begin{aligned}
f_{qk} &= \text{MLP}(l_{qk}, d_x, d_h, d_h) \\
\mathbf{Q} &= f_{qk}(\mathbf{x}_i), \quad i \in \mathcal{T} \\
\mathbf{K} &= \{f_{qk}(\mathbf{x}_i)\}, \quad i \in \mathcal{C} \\
\mathbf{V} &= \text{SA}(d_h).(\{\text{MLP}(l_v, d_x + d_y, d_h, d_h)([\mathbf{x}_i, y_i])\}_{i \in \mathcal{C}}) \\
\phi_1 &= \text{MHA}(d_h)(\mathbf{Q}, \mathbf{K}, \mathbf{V}) \\
\mathbf{H} &= \text{SA}(d_h)\left(\{\text{RELU} \circ \text{MLP}(l_{e1}, d_x + d_y, d_h, d_h)([\mathbf{x}_i, y_i])\}_{i \in \mathcal{C}}\right) \\
\phi_2 &= \text{MLP}(l_e, d_h, d_h)\left(\frac{1}{|\mathcal{C}|}\sum_{i \in \mathcal{C}}\mathbf{h}_i\right) \\
\phi &= [\phi_1, \phi_2]
\end{aligned}
\tag{36}
$$

Similarly, an encoder with attention in NP contains a deterministic path and a latent path, i.e.,

$$
\begin{aligned}
f_{qk} &= \text{MLP}(l_{qk}, d_x, d_h, d_h) \\
\mathbf{Q} &= f_{qk}(\mathbf{x}_i), \quad i \in \mathcal{T} \\
\mathbf{K} &= \{f_{qk}(\mathbf{x}_i)\}, \quad i \in \mathcal{C} \\
\mathbf{V} &= \text{SA}(d_h).(\{\text{MLP}(l_v, d_x + d_y, d_h, d_h)([\mathbf{x}_i, y_i])\}_{i \in \mathcal{C}}) \\
\phi &= \text{MHA}(d_h)(\mathbf{Q}, \mathbf{K}, \mathbf{V})
\end{aligned}
\tag{37}
$$

and

$$
\begin{aligned}
\mathbf{H} &= \text{SA}(d_h)\left(\{\text{RELU} \circ \text{MLP}(l_{e1}, d_x + d_y, d_h, d_h)([\mathbf{x}_i, y_i])\}_{i \in \mathcal{C}}\right) \\
[\mu_z, \sigma'_z] &= \text{MLP}(l_{la}, d_h, d_h)\left(\frac{1}{|\mathcal{C}|}\sum_{i \in \mathcal{C}}\mathbf{h}_i\right) \\
\sigma_z &= 0.1 + 0.9 \cdot \text{SIGMOID}(\sigma'_z), \\
\mathbf{z} &\sim \mathcal{N}(\mu_z, \text{diag}(\sigma_z^2)).
\end{aligned}
\tag{38}
$$

## C.3. Decoder

The decoder focuses on predicting output for target points based on the encoder's outputs $\phi$. For target point $\{\mathbf{x}_i\}_{i \in \mathcal{T}}$, the decoder of CNP is defined by

$$
\begin{aligned}
[\mu_i, \sigma_i'] &= \text{MLP}(d_{dec}, 2d_h + d_x, d_h, 2d_y)[\phi, \mathbf{x}_i], \quad i \in \mathcal{T} \\
\sigma_i &= 0.1 + 0.9 \cdot \text{SOFTPLUS}(\sigma_i') \\
y_i &\sim \mathcal{N}(\mu_i, \sigma_i)
\end{aligned}
\tag{39}
$$

Decoder of NP is defined by

$$
\begin{aligned}
[\mu_i, \sigma_i'] &= \text{MLP}(d_{dec}, d_h + d_z + d_x, d_h, 2d_y)[\phi, \mathbf{x}_i, \mathbf{z}], \quad i \in \mathcal{T} \\
\sigma_i &= 0.1 + 0.9 \cdot \text{SOFTPLUS}(\sigma_i') \\
y_i &\sim \mathcal{N}(\mu_i, \sigma_i)
\end{aligned}
\tag{40}
$$

# D. Implementation Details and Experiments

## D.1. Infrastructure

We implement our model with Pytorch, and conduct our experiments with:

- CPU: Intel Xeon Silver 4316.

- GPU: 8x GeForce RTX 4090.

- RAM: DDR4 384GB.

- ROM: 16TB 7.2K 6Gb SATA and 1x 960G SATA 6Gb R SSD.

- Operating system: Ubuntu 18.04 LTS.

- Environments: Python 3.7; NumPy 1.18.1; SciPy 1.2.1; scikit-learn 0.23.2; seabornn 0.1; torch_geometric 1.6.1; matplotlib 3.1.3; dgl 0.4.2; pytorch 1.6.

## D.2. Experimental Settings

For CNP (Garnelo et al., 2018a), BCNP (Lee et al., 2020), SCNP (Liu et al., 2024a), and our RCNP, we apply the encoder with attention described in Eq (36) and decoder described in Eq (39). For NP (Garnelo et al., 2018b), ANP (Kim et al., 2019), BNP (Lee et al., 2020), BANP (Lee et al., 2020) and our SNP and SANP models, we apply encoder with attention described in Eq (37) and (38), and decoder described in Eq (40).

### D.2.1. 1D REGRESSION

For synthetic 1D regression experiments, the neural architectures for CNP, NP, ANP, BCNP, BNP, BANP, and our SCNP/SNP/SANP refer to Appendix C. The number of hidden units is $d_h = 128$ and latent representation $d_z = 128$. The number of layers are $l_e = l_{de} = l_{la} = l_{qk} = l_v = 2$.

We generate datasets for synthetic 1D regression. Specifically, the stochastic process (SP) initializes with a 0 mean Gaussian Process (GP) $y^{(0)} \sim GP(0, k(\cdot, \cdot))$ indexed in the interval $x \in [-2.0, 2.0]$, where the radial basis function kernel $k(x, x') = \sigma^2 \exp(-\|x - x'\|^2/2l^2)$ with $s \sim U(0.1, 1.0)$ and $\sigma \sim U(0.1, 0.6)$. Furthermore, GP with Matern Kernel is adopted for model-data mismatch scenario, which is defined as $k(x, x') = \sigma^2(1 + \sqrt{5}d/l + 5d^2/(3l^2)) \exp(-\sqrt{5}d/l)$ and $d = \|x - x'\|$ with $s \sim U(0.1, 1.0)$ and $\sigma \sim U(0.1, 0.6)$. For a fair comparison, we set the same data generation, training, and testing for all models.

We trained all models for $100,000$ steps with each step computing updates with a batch containing 100 tasks. We used the Adam optimizer with an initial learning rate $5 \cdot 10^{-4}$ and decayed the learning rate using Cosine annealing scheme for baselines. For SCNP/SNP/SANP, we set $K = 3$. The size of the context $\mathcal{C}$ was drawn as $|\mathcal{C}| \sim U(3, 200)$. Testings were done for $3,000$ batches with each batch containing 16 tasks ($48,000$ tasks in total).

### D.2.2. IMAGE COMPLETION

Analogous to the 1D experiments, we take random pixels of a given image at training as targets, and select a subset of this as contexts, again choosing the number of contexts and targets randomly ($n \sim U[3, 200]$, $m \sim n + U[0, 200 - n]$). The $\mathbf{x}_i$ are rescaled to $[-1, 1]$ and the $y_i$ are rescaled to $[-0.5, 0.5]$. We use a batch size of 16 for both EMNIST and CelebA, i.e. use 16 randomly selected images for each batch.

For image completion experiments on EMNIST and CelebA dataset, the neural architectures for CNP, NP, ANP, BCNP, BNP, BANP, and our SCNP, SNP, and SANP refer to Appendix C. The number of hidden unites is $d_h = 128$ and latent representation $d_z = 128$. The number of layers are $l_e = l_{de} = 4, l_{la} = l_{qk} = l_v = 5$. $h_{head} = 8$

### D.2.3. BAYESIAN OPTIMIZATION

We sampled 100 GP prior functions from zero mean and unit variance. After realizing them, the prior functions are used to optimize via Bayesian optimization. We normalized these functions in order to fairly compare simple regrets and cumulative regrets across distinct sampled functions (Basically, since they are sampled from the same distributions, the scales of them are quite similar, but we used more precise evaluations).

As presented in the Bayesian optimization results, all the methods are started from the same initializations. We employed Gaussian process regression (Rasmussen, 2003) with squared exponential kernels as a surrogate model, and expected improvement (Jones et al., 1998) as an acquisition function, which is optimized by the multi-started local optimization method, L-BFGS-B with 100 initial points.

### D.2.4. CONXTUAL BANDITS

The whell bandit problem is introduced in (Riquelme et al., 2018) and can be illustrated in Figure 5. In this problem, a unit circle is divided into a low-reward region (blue area) and four high-reward regions (the other four colored areas). A scalar $\gamma$ determines the size of the low-reward region, and other regions have equal sizes. The agent does not know the underlying $\gamma$, and has to choose among $k = 5$ arms given its coordinates $X = (X_1, X_2)$ within the circle. If $\|X\| \le \gamma$, the agent falls within the low-reward region (blue). In this case, the optimal action is $k = 1$, which provides a reward $r \sim N(1.2, 0.012)$, while all other actions only return $r \sim N(1.0, 0.012)$. If the agent falls within any of the four high-reward regions ($\|X\| > \gamma$), the optimal arm will be one of the remaining four $k = 2 - 5$, depending on the specific area. Pulling the optimal arm here results in a high reward $r \sim N(50.0, 0.012)$, and as before all other arms receive $N(1.0, 0.012)$ except for arm $k = 1$ which always returns $N(1.2, 0.012)$.

We sample a dataset of $K$ different wheel problems $\gamma_i{}_{i=1}^K$, which are drawn from a uniform distribution $\gamma \sim U(0, 1)$. For each problem, we sample $N$ points to evaluate and pick m points as context, in which each point is a tuple $(X, r)$ of the coordinates $X$ and the corresponding reward values $r$ of all 5 arms. The training objective is to regress the reward values from the coordinates. We set $K = 8, N = 562, m = 512$ in our experiments.

## E. Additional Experiments

### E.1. 1D Regression

Following (Kim et al., 2019; Lee et al., 2020; Liu et al., 2024a), we first conduct 1D regression experiments. We generate training data from a GP with RBF kernels and test the trained model on the data generated from GPs with different kernels, including RBF, Matérn, and Periodic. Among them, data from Matérn and Periodic are used to evaluate model performance on unseen functions. We give a detailed description of experimental settings in the Appendix. Table 4 lists the average log-likelihood comparison in terms of different kernels. The best results are marked in bold. After several repetitive observations, we can observe that our robust version outperforms not only vanilla ones but also bootstrapping versions and stable versions. For model-data mismatch settings, all the models become less accurate, but our robust ones affect less. Visualizations for 1D regression data are shown in Figure 6. Each solid blue curve represents one sample function and the blue area around each curve represents the variance in the predictive distribution. We can see that NPs with bootstraping, stable, and robust solutions generally achieve better fitting performance, especially on data points where other methods fail to fit effectively.

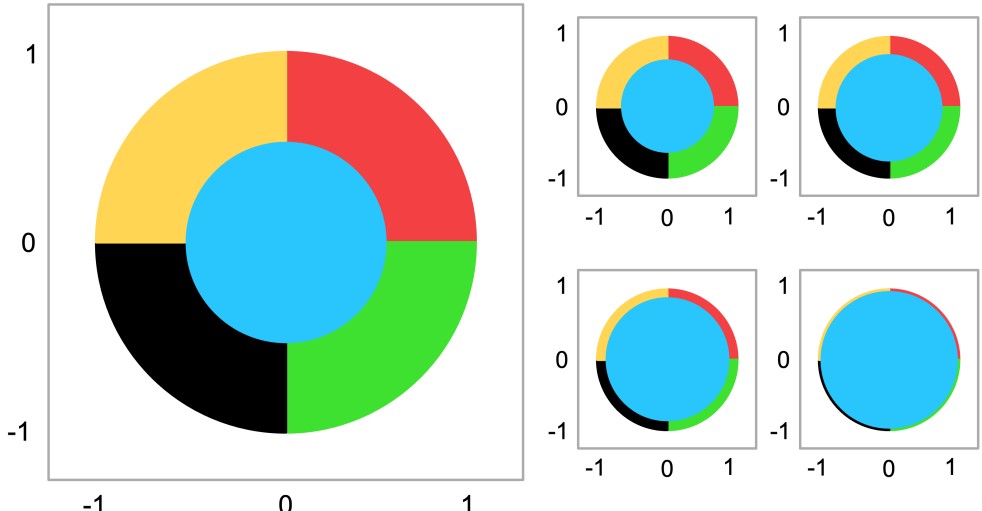

*Figure 5.* The wheel bandit problem with varying values of $\gamma$. (Riquelme et al., 2018; Nguyen & Grover, 2022).

*Table 4.* Comparison of our robust NPs with the baselines on log-likelihood of the target points on various GP kernels. We train each method with 5 different seeds and report the mean and standard deviation.

| Method | RBF | Matérn 5/2 | Periodic |
|---|---|---|---|
| CNP | $0.4334_{\pm 0.007}$ | $0.2431_{\pm 0.010}$ | $-1.2521_{\pm 0.008}$ |
| BCNP | $0.4589_{\pm 0.006}$ | $0.2762_{\pm 0.009}$ | $-1.1915_{\pm 0.007}$ |
| SCNP | $0.4733_{\pm 0.004}$ | $0.2953_{\pm 0.006}$ | $-1.1779_{\pm 0.006}$ |
| RCNP | $\mathbf{0.4798}_{\pm 0.003}$ | $\mathbf{0.3021}_{\pm 0.005}$ | $\mathbf{-1.1699}_{\pm 0.006}$ |
| NP | $0.3853_{\pm 0.005}$ | $0.2041_{\pm 0.015}$ | $-1.4255_{\pm 0.009}$ |
| BNP | $0.4211_{\pm 0.004}$ | $0.2689_{\pm 0.007}$ | $-1.3988_{\pm 0.009}$ |
| SNP | $0.4356_{\pm 0.004}$ | $\mathbf{0.2844}_{\pm 0.005}$ | $-1.4085_{\pm 0.008}$ |
| RNP | $\mathbf{0.4398}_{\pm 0.002}$ | $0.2832_{\pm 0.006}$ | $\mathbf{-1.3761}_{\pm 0.009}$ |
| ANP | $0.5763_{\pm 0.004}$ | $0.6366_{\pm 0.004}$ | $-1.1824_{\pm 0.009}$ |
| BANP | $0.5887_{\pm 0.006}$ | $0.6514_{\pm 0.005}$ | $-1.1781_{\pm 0.006}$ |
| SANP | $0.5994_{\pm 0.004}$ | $0.6653_{\pm 0.004}$ | $\mathbf{-1.1774}_{\pm 0.004}$ |
| RANP | $\mathbf{0.6025}_{\pm 0.002}$ | $\mathbf{0.6771}_{\pm 0.006}$ | $-1.1779_{\pm 0.006}$ |
| ConvNP | $0.6503_{\pm 0.004}$ | $0.6557_{\pm 0.005}$ | $-1.1761_{\pm 0.006}$ |
| BConvNP | $0.6531_{\pm 0.005}$ | $0.6787_{\pm 0.006}$ | $-1.1675_{\pm 0.006}$ |
| SConvNP | $0.6831_{\pm 0.003}$ | $0.6836_{\pm 0.004}$ | $-1.1681_{\pm 0.008}$ |
| RConvNP | $\mathbf{0.6898}_{\pm 0.003}$ | $\mathbf{0.6916}_{\pm 0.004}$ | $\mathbf{-1.1547}_{\pm 0.004}$ |
| TNP | $0.8711_{\pm 0.003}$ | $0.7151_{\pm 0.001}$ | $-1.1625_{\pm 0.005}$ |
| BTNP | $\mathbf{0.8981}_{\pm 0.006}$ | $0.7255_{\pm 0.003}$ | $-1.1591_{\pm 0.008}$ |
| STNP | $0.8798_{\pm 0.008}$ | $0.7151_{\pm 0.006}$ | $-1.1552_{\pm 0.002}$ |
| RTNP | $0.8815_{\pm 0.004}$ | $\mathbf{0.7389}_{\pm 0.003}$ | $\mathbf{-1.1511}_{\pm 0.004}$ |

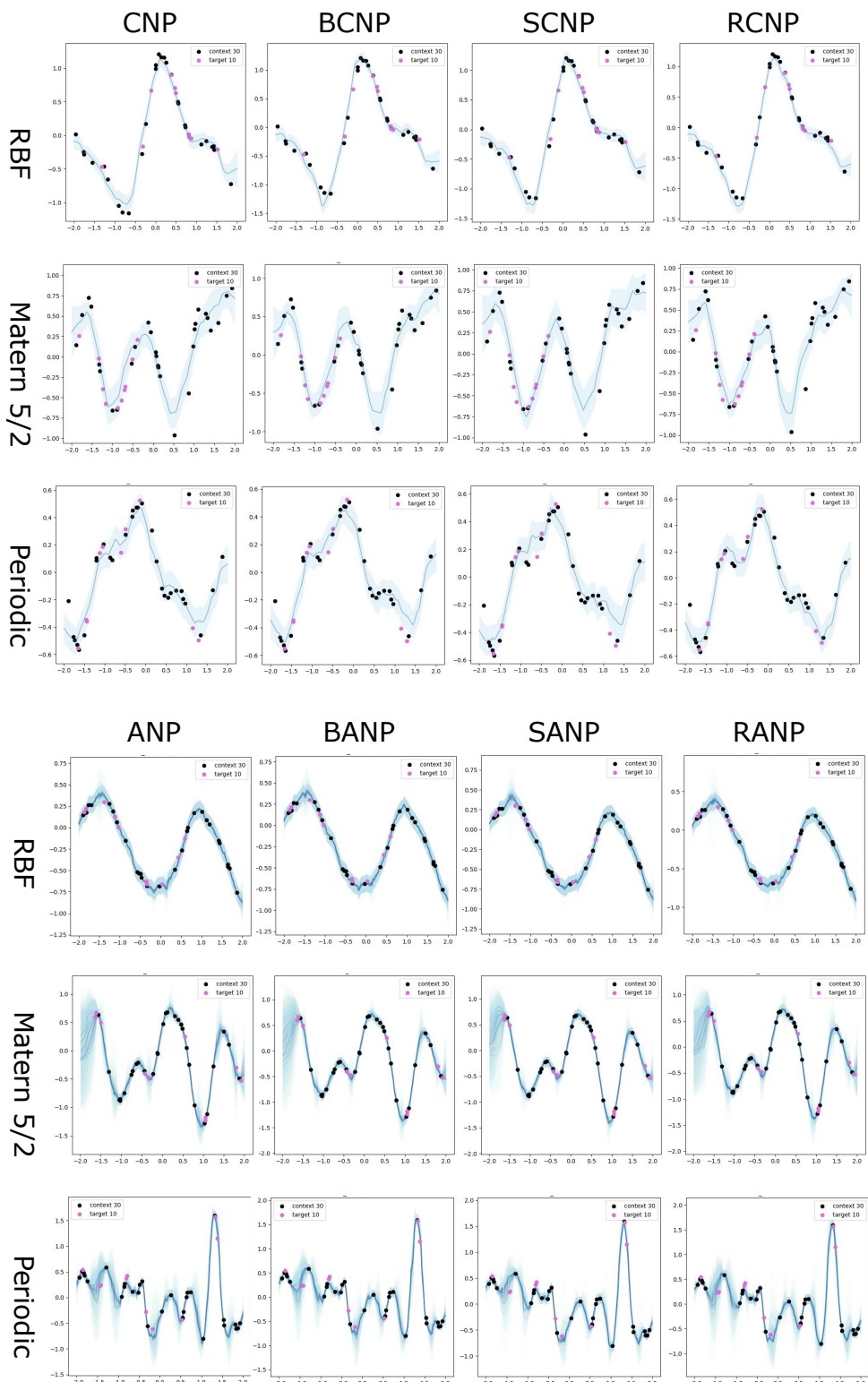

*Figure 6.* Visualizations of CNP, ANP, and their bootstrap versions (BCNP, BANP), stable versions (SNP, SANP), and robust versions (RCNP, RANP) for 1D regression data.

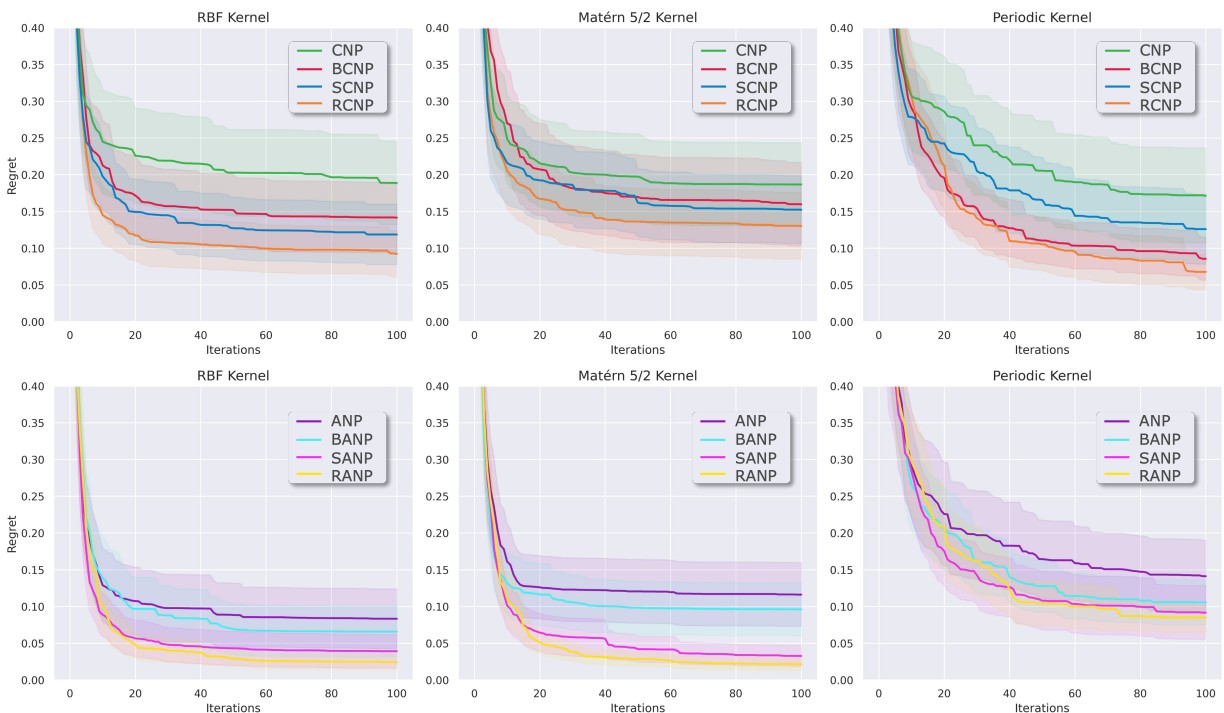

*Figure 7.* Regret performance on 1D Bayesian optimization tasks. For each kernel, we generate 100 functions and report the mean and standard deviation.

### E.2. Bayesian Optimization

Taking GP data with RBF, Matérn 5/2, and periodic prior functions as examples, we gave the results of CNP, ANP, corresponding bootstrapping NPs (BCNP, BANP), stable NPs (SCNP, SANP), and robust NPs (RCNP, RANP). Note that we use NPs as the surrogate functions and Upper Confidence Bound (UCB) as the acquisition function. To maintain consistent comparison, we standardized the initializations and normalized the results. For each objective function, we run Bayesian optimization for 100 iterations, and simple regret is used as the evaluation metric. As shown in Figure 7 and Table 5, we can see that our robust solutions consistently achieve lower regret than other NPs in all three kernels.

### E.3. The effect of confidence level $\alpha$

The key parameter in our robust solution is the confidence level $\alpha$, which controls the extent to which the learning model focuses on difficult tasks. Taking RANP as an example, we investigate the average log-likelihood in terms of different $\alpha$ on the 1D regression, image completion, and Bayesian optimization, as shown in Figure 8. We can see that RANP obtains the best results (around $0.6$), after that, the performance remained stable. This result is in line with our intuition. If there is too much focus on difficult tasks, it may neglect the learning of general tasks. On the other hand, insufficient focus on difficult tasks can also affect the model's performance.

### E.4. Risk Distribution

As we stated previously, the proposed strategy can iteratively reshape the task risk distribution to increase robustness, i.e., transport the probability mass in high-risk regions to the lower-risk regions. To verify this point, we present the task risk distributions with and without using our strategy. Specifically, taking RANP as an example, Figure 9 shows the task risk distributions of ANP and RANP on, 1d regression, image completion, and Bayesian optimization tasks. We select MSE as the risk measure. Overall, it can be seen that the robust solutions shift the original risk distribution to the left, reducing the proportion of high-risk tasks and effectively demonstrating the robustness of the proposed model.

*Table 5.* Bayesian optimization experiments on data generated by different GP kernels

| Method | RBF | Matérn 5/2 | Periodic |
|--------|-----|------------|----------|
| CNP | $0.1524_{\pm 0.004}$ | $0.1852_{\pm 0.004}$ | $0.2051_{\pm 0.005}$ |
| BCNP | $0.1501_{\pm 0.003}$ | $0.1815_{\pm 0.003}$ | $0.2016_{\pm 0.003}$ |
| SCNP | $0.1415_{\pm 0.005}$ | $0.1752_{\pm 0.003}$ | $0.1998_{\pm 0.003}$ |
| RCNP | $\mathbf{0.1352}_{\pm 0.003}$ | $\mathbf{0.1711}_{\pm 0.004}$ | $\mathbf{0.1941}_{\pm 0.004}$ |
| NP | $0.1647_{\pm 0.003}$ | $0.1988_{\pm 0.003}$ | $0.1985_{\pm 0.007}$ |
| BNP | $0.1611_{\pm 0.003}$ | $0.1901_{\pm 0.004}$ | $0.1962_{\pm 0.003}$ |
| SNP | $0.1536_{\pm 0.004}$ | $0.1871_{\pm 0.003}$ | $0.1915_{\pm 0.005}$ |
| RNP | $\mathbf{0.1415}_{\pm 0.003}$ | $\mathbf{0.1789}_{\pm 0.005}$ | $\mathbf{0.1873}_{\pm 0.003}$ |
| ANP | $0.1245_{\pm 0.003}$ | $0.1518_{\pm 0.003}$ | $0.1892_{\pm 0.002}$ |
| BANP | $0.1341_{\pm 0.003}$ | $0.1316_{\pm 0.004}$ | $0.1788_{\pm 0.005}$ |
| SANP | $0.1142_{\pm 0.002}$ | $0.1201_{\pm 0.002}$ | $\mathbf{0.1672}_{\pm 0.001}$ |
| RANP | $\mathbf{0.1025}_{\pm 0.002}$ | $\mathbf{0.1171}_{\pm 0.003}$ | $0.1779_{\pm 0.006}$ |
| ConvNP | $0.1211_{\pm 0.004}$ | $0.1256_{\pm 0.003}$ | $0.1854_{\pm 0.004}$ |
| BConvNP | $0.1187_{\pm 0.003}$ | $0.1201_{\pm 0.004}$ | $0.1819_{\pm 0.003}$ |
| SConvNP | $0.1176_{\pm 0.006}$ | $0.1179_{\pm 0.003}$ | $0.1786_{\pm 0.004}$ |
| RConvNP | $\mathbf{0.1056}_{\pm 0.004}$ | $\mathbf{0.1156}_{\pm 0.004}$ | $\mathbf{0.1773}_{\pm 0.003}$ |
| TNP | $0.1125_{\pm 0.003}$ | $0.1451_{\pm 0.001}$ | $0.1715_{\pm 0.003}$ |
| BTNP | $0.1037_{\pm 0.006}$ | $0.1455_{\pm 0.003}$ | $0.1691_{\pm 0.008}$ |
| STNP | $0.0998_{\pm 0.004}$ | $0.1351_{\pm 0.006}$ | $0.1561_{\pm 0.002}$ |
| RTNP | $\mathbf{0.0915}_{\pm 0.004}$ | $\mathbf{0.1289}_{\pm 0.003}$ | $\mathbf{0.1463}_{\pm 0.004}$ |

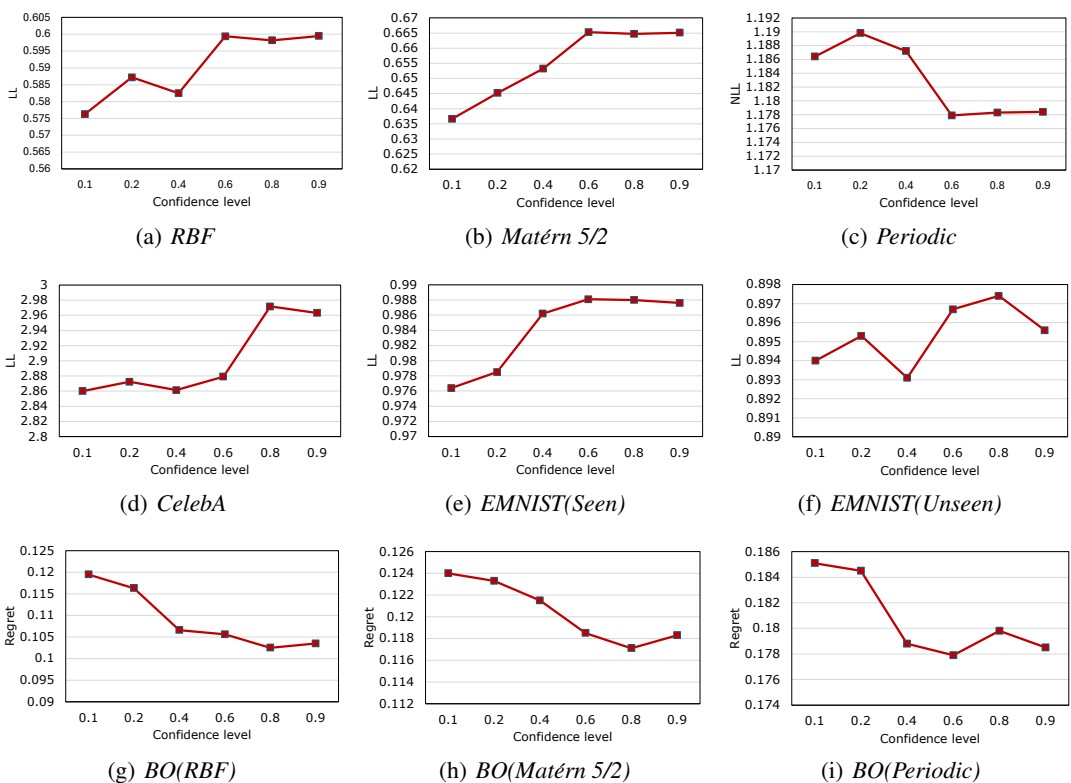

(a) *RBF*  (b) *Matérn 5/2*  (c) *Periodic*

(d) *CelebA*  (e) *EMNIST(Seen)*  (f) *EMNIST(Unseen)*

(g) *BO(RBF)*  (h) *BO(Matérn 5/2)*  (i) *BO(Periodic)*

*Figure 8.* The performance evaluation when we set the confidence level $\alpha$ with different values.

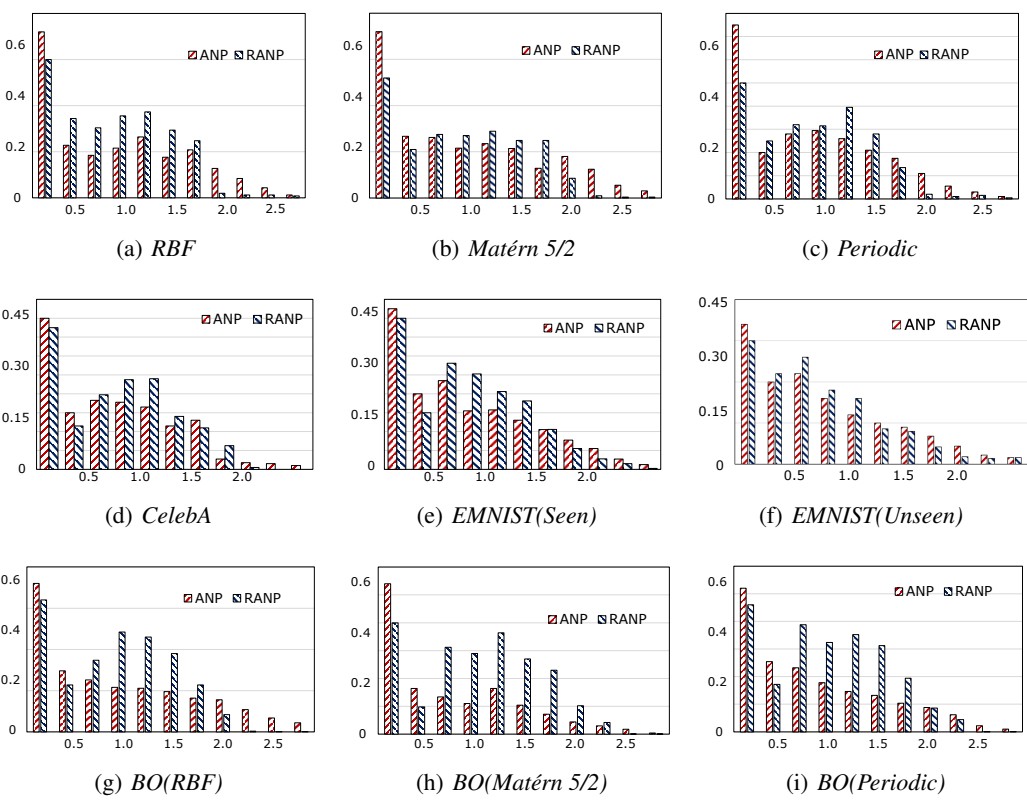

*Figure 9.* The histograms of task risks in different tasks.

*Table 6.* Comparison of our robust NPs under different weight strategies on log-likelihood of the target points on various GP kernels. We train each method with 5 different seeds and report the mean and standard deviation.

| Method | RBF | | Matérn 5/2 | | Periodic | |
|---|---|---|---|---|---|---|
| | average | adam | average | adam | average | adam |
| RCNP | $0.4756_{\pm 0.002}$ | $0.4798_{\pm 0.003}$ | $0.2989_{\pm 0.003}$ | $0.3021_{\pm 0.005}$ | $-1.1751_{\pm 0.005}$ | $-1.1699_{\pm 0.006}$ |
| RNP | $0.4359_{\pm 0.003}$ | $0.4398_{\pm 0.002}$ | $0.2797_{\pm 0.003}$ | $0.2832_{\pm 0.006}$ | $-1.3895_{\pm 0.007}$ | $-1.3761_{\pm 0.009}$ |
| RANP | $0.5989_{\pm 0.004}$ | $0.6025_{\pm 0.002}$ | $0.6688_{\pm 0.005}$ | $0.6771_{\pm 0.006}$ | $-1.1797_{\pm 0.007}$ | $-1.1779_{\pm 0.006}$ |
| RConvNP | $0.6857_{\pm 0.004}$ | $0.6898_{\pm 0.003}$ | $0.6871_{\pm 0.004}$ | $0.6916_{\pm 0.004}$ | $-1.1699_{\pm 0.008}$ | $-1.1547_{\pm 0.004}$ |
| RTNP | $0.8796_{\pm 0.005}$ | $0.8815_{\pm 0.004}$ | $0.7276_{\pm 0.004}$ | $0.7389_{\pm 0.003}$ | $-1.1544_{\pm 0.006}$ | $-1.1511_{\pm 0.004}$ |

*Table 7.* Comparison of our robust NPs under different weight strategies on log-likelihood of the target points on two real-world datasets: CelebA and EMNIST. We train each method with 5 different seeds and report the mean and standard deviation.

| Method | CelebA | | EMNIST(Seen (0-9)) | | EMNIST(Unseen (10-46)) | |
|---|---|---|---|---|---|---|
| | average | adam | average | adam | average | adam |
| RCNP | $2.1735_{\pm 0.007}$ | $2.1896_{\pm 0.005}$ | $0.7845_{\pm 0.005}$ | $0.7879_{\pm 0.005}$ | $0.5321_{\pm 0.005}$ | $0.5362_{\pm 0.006}$ |
| RNP | $2.8871_{\pm 0.006}$ | $2.8915_{\pm 0.006}$ | $0.8876_{\pm 0.004}$ | $0.8911_{\pm 0.005}$ | $0.7101_{\pm 0.006}$ | $0.7121_{\pm 0.006}$ |
| RANP | $2.9651_{\pm 0.003}$ | $2.9718_{\pm 0.006}$ | $0.9825_{\pm 0.006}$ | $0.9881_{\pm 0.004}$ | $0.8915_{\pm 0.007}$ | $0.8967_{\pm 0.005}$ |
| RConvNP | $3.2215_{\pm 0.004}$ | $3.2319_{\pm 0.005}$ | $1.2251_{\pm 0.006}$ | $1.2348_{\pm 0.005}$ | $1.0663_{\pm 0.005}$ | $1.0721_{\pm 0.006}$ |
| RTNP | $4.4189_{\pm 0.003}$ | $4.4226_{\pm 0.005}$ | $1.5541_{\pm 0.003}$ | $1.5572_{\pm 0.002}$ | $1.4361_{\pm 0.005}$ | $1.4413_{\pm 0.004}$ |

## E.5. The effect of $w_k$

In the gradient computation process of the outer loop, the weighting coefficient for the gradient from the previous step is typically determined using an average weighting strategy, i.e., $w_{k-1}^{s-1} = \frac{1}{K_{s-1}}, \forall k = 1, \cdots, K_{s-1}$. However, we can set it as a learning rate $\eta_k^s$ as shown in Eq. (15). We compared the performance of the robust model using the average strategy and adaptive learning rates, using three tasks as examples. The results are shown in Table 6, 7, and 8. It can be observed that the experimental results with the adaptive learning rate (Adam) outperformed the average strategy, while the average strategy performed better than other bootstrap strategies or the stable strategy.

*Table 8.* Comparison of our robust NPs under different weight strategies on Bayesian optimization experiments with different GP kernels. We train each method with 5 different seeds and report the mean and standard deviation.

| Method | RBF | | Matérn 5/2 | | Periodic | |
|---|---|---|---|---|---|---|
| | average | adam | average | adam | average | adam |
| RCNP | $0.1396_{\pm 0.003}$ | $0.1352_{\pm 0.003}$ | $0.1753_{\pm 0.003}$ | $0.1711_{\pm 0.004}$ | $0.1985_{\pm 0.005}$ | $0.1941_{\pm 0.004}$ |
| RNP | $0.1489_{\pm 0.004}$ | $0.1415_{\pm 0.003}$ | $0.1837_{\pm 0.003}$ | $0.1789_{\pm 0.005}$ | $0.1908_{\pm 0.004}$ | $0.1873_{\pm 0.003}$ |
| RANP | $0.1097_{\pm 0.003}$ | $0.1025_{\pm 0.002}$ | $0.1204_{\pm 0.003}$ | $0.1171_{\pm 0.003}$ | $0.1798_{\pm 0.005}$ | $0.1779_{\pm 0.006}$ |
| RConvNP | $0.1128_{\pm 0.005}$ | $0.1056_{\pm 0.004}$ | $0.1180_{\pm 0.004}$ | $0.1156_{\pm 0.004}$ | $0.1793_{\pm 0.006}$ | $0.1773_{\pm 0.003}$ |
| RTNP | $0.0973_{\pm 0.001}$ | $0.0915_{\pm 0.004}$ | $0.1327_{\pm 0.005}$ | $0.1289_{\pm 0.003}$ | $0.1517_{\pm 0.006}$ | $0.1463_{\pm 0.004}$ |

