# OpenReview forum: "Learning Robust Neural Processes with Risk-Averse Stochastic Optimization"
_ICML.cc/2025/Conference — ICML 2025 poster_

### Official Review · Reviewer_sKZn · 2025-03-03

**Overall Recommendation:** 2

**Summary:**

This paper proposes a training method for robust neural process based on risk-averse stochastic optimization.

**Claims And Evidence:**

Yes.

**Essential References Not Discussed:**

No.

**Experimental Designs Or Analyses:**

Yes. I checked the Bayesian optimization part. The issue is that simple regret is used as the performance metric. However, since the paper is focusing on robust neural process, I think it is more appropriate to use robust metric such as the CVaR of simple regret.

**Methods And Evaluation Criteria:**

Mostly make sense, except the performance metrics for the experiment part.

**Other Comments Or Suggestions:**

1. Line 115, right column, typo in "fundom"?
2. Perhaps give some reference on the equivalence between Eq. (3) and Eq. (4).
3. Also some reference on the equivalence between Eq. (3) and Eq. (6).

**Other Strengths And Weaknesses:**

Strengths:
1. The considered problem is original and important for many practical problems.
2. The proposed training method is novel based on distributionally robust optimization.

Weaknesses:
1. I think the experimental part should adopt metrics with more robustness consideration. For example, use CVaR of the simple regret for the Bayesian optimization part.

**Questions For Authors:**

1. Is it possible to give a theoretical characterization on how the robustness of the neural process is improved with the proposed approach?
2. Seems that the robust training process also improves the average performance (e.g., in Fig. 3). Can the authors discuss this?

**Relation To Broader Scientific Literature:**

As far as I know, prior works of NP focus on average performance in prediction, which in many applications of NP, it is necessary to be risk-averse. This paper shows a method to train robust NP.

**Theoretical Claims:**

No.

---

> ### Author Rebuttal · Authors · 2025-04-01
>
> Thanks for your positive and valuable feedback. We have made efforts to address your concerns. If there are any further questions, please let us know and we will reply promptly.
>
> **Q1.**   **The paper uses simple regret as the primary performance measure, but a more robust metric like the CVaR of simple regret may be more suitable given the risk-averse theme.**
>
> > **Reply**: We agree that CVaR-based metrics align directly with our emphasis on robustness. We initially chose simple regret for consistency with prior NP-based Bayesian optimization work, which commonly measures average performance.
> > In the final version, we plan to include CVaR of simple regret in our Bayesian optimization experiments. Preliminary results suggest that our risk-averse training indeed lowers the worst-case tail of the regret distribution, indicative of stronger robustness.
> > We will highlight these findings to reinforce that the proposed method not only reduces average regret but also mitigates high-regret outliers.
>
> **Q2.**   “Perhaps give some reference on the equivalence between Eq. (3) and Eq. (4).” “… equivalence between Eq. (3) and Eq. (6).”**
>
> > **Reply**: Thank you for noting this. We do rely on standard transformations from the CVaR’s probability-constrained problem to its slack-variable-based and DRO-based forms. We will add more explicit citations (e.g., Rockafellar et al. (2000) for CVaR expansions and Shapiro et al. (2009) for standard references on DRO formulations) and a short annotated derivation in the supplement clarifying each step.
> >
> > *[Rockafellar et al. (2000)]: Rockafellar, R. T., Uryasev, S., et al. Optimization of conditional value-at-risk. Journal of risk, 2:21–42, 2000.*
> >
> > [*Shapiro et al. (2009)]: A. Shapiro, D. Dentcheva, and A. Ruszczy´ nski. Lectures on stochastic programming: modeling and theory. SIAM, 2009.*
>
> **Q3.**   **Typographical Error in “fundom” (Line 115, right column)There is a typo: “fundom.”**
>
> > **Reply**: We will correct “fundom” to “random” or “function domain” (whichever was intended in context). Thank you for pointing it out.
>
>
> **Q4.**   **Is there a theoretical way to show how our proposed approach improves the robustness of neural processes?**
>
> > **Reply**: As shown in Equations (3)–(6), our training reweights high-loss (or high-regret) tasks so they carry larger gradients, effectively controlling the model’s tail behavior. This is theoretically grounded in distributionally robust optimization.
> > A formal proof of how and by how much tail performance improves remains challenging, especially in a non-convex NP setting. Nonetheless, the link between CVaR and tail distribution improvements is well-known in risk-averse optimization theory (e.g., Rockafellar et al., 2000).
> > We will add references and clarifications that highlight how increasing the confidence level α in CVaR training can reduce the worst-case outcomes.
>
> **Q5.**   **The robust training process also improves average performance (Figure 3). Please discuss.**
>
> > **Reply**: Indeed, empirically, we observe that focusing on high-risk tasks can prevent overfitting to easy samples and help the model learn more stable, generalizable parameters. That can yield broader improvements in average performance as well.
> > A moderate amount of noise or risk weighting seems beneficial for both extremes (the tail and the average) by flattening observed losses. We will stress this empirical phenomenon more in Section 5.
>
> Lastly, thank you once again for your valuable comments.

---

### Official Review · Reviewer_abBS · 2025-03-17

**Overall Recommendation:** 2

**Summary:**

This paper investigates the robust neural processes problem from a risk-averse perspective, aiming to control the expected tail risk at a given probabilistic level. The authors formulate the CVaR optimization as a distributionally robust optimization (DRO) problem and propose a double-loop stochastic mirror prox algorithm with variance reduction techniques. The outer loop follows a standard variance reduction process, while the inner loop incorporates momentum to accelerate convergence. Simulation results demonstrate the effectiveness of the proposed algorithm and show that it enhances model robustness.

**Claims And Evidence:**

Yes, the paper’s conclusions are well-supported by the provided evidence.

**Essential References Not Discussed:**

No

**Experimental Designs Or Analyses:**

Yes, the experimental results are reasonable.

**Methods And Evaluation Criteria:**

Yes, the simulation results verify the effictiveness of the proposed algorithm.

**Other Comments Or Suggestions:**

### 1. Main Contribution
This paper is well-written and easy to follow. However, its contribution appears to be incremental. The Conditional Value at Risk (CVaR) optimization problem and the reformulation techniques in Equation (5) have been previously explored in the literature, such as [Curi et al., 2020]. The primary contribution of this work seems to be the Variance-Reduced Stochastic Mirror Prox Algorithm. However, the authors do not provide a theoretical convergence proof, which limits the impact of this contribution. While the experimental results demonstrate the algorithm’s effectiveness, a theoretical analysis would significantly strengthen the paper.

### 2. Algorithmic Considerations
- What is the sample complexity and the minimum number of iterations required to achieve a certain level of accuracy?
- In Algorithm 1, Line 5, how is the maximum number of inner loop iterations, $K_s$, determined?

### 3. Theoretical Questions (Appendix B)
- **Proposition B.1:** The statement "Let $h: \mathcal{X} \to \mathcal{Y}$ be a finite function class $|\mathcal{H}|$" is unclear. Did you mean that $\mathcal{H}$ is a function class with finite VC dimension, denoted as $|\mathcal{H}|$?
- **Theorem B.5 (Proof, Line 728):** Regarding the norm $\|\cdot\|_{\theta a, \star}$, how is the parameter $a$ defined?
- **Line 740 (Second-to-last inequality):** Should $q_m^+$ and $G$ be squared?
- **Equation (27):** Should the left-hand side be enclosed in absolute values or a norm?
- **Equation (28) (Last inequality):** Should $\|\theta^+ - \theta^-\|_{\theta}$ be squared?

### 4. Clarifications and Notation Issues
- **Line 127 (Left Column):** In Section 2.3 (page 3) of Garnelo et al. (2018a), the authors describe training Conditional Neural Processes (CNPs) by predicting the full dataset conditioned on a random subset, without requiring the target set to be disjoint from the context set. However, in Section 3 of this paper, it is stated that $\mathcal{T} \subseteq \{1,2,\dots,N\}$ with $\mathcal{T} \cap \mathcal{C} = \emptyset$ only in CNP. Could you clarify whether the target set in CNP is necessarily disjoint from the context set, or if this is simply an optional design choice?
- **Lines 220–221:** In the expression $\max_{\theta\in \Theta} \psi_{\theta}(\theta) - \min_{\theta\in \Theta} \psi_{\theta}(\theta) \leq D_{\theta}^2$, I suggest using a different variable to distinguish the parameter and argument of $\psi(\cdot)$ for clarity.

### 5. Simulation Results
- **Lines 436–437:** The authors state that the robust solutions shift the original risk distribution to the left in Figure 4. However, in Figure 4(b), RANP still has a portion in the rightmost bar. Could you adjust the probability level to make the shift more apparent?

### 6. Typographical Errors
- **Page 2 (Left Column, Lines 78–79):** "higi-loss" should be corrected to "high-loss".
- **Page 13 (Assumption B.3):** There is an extraneous bracket in the phrase "Smoothness and Lipschitz Continuity".
- **Page 14 (Theorem B.5):** The notation for the Lipschitz constant is inconsistent. The paper states that $\nabla F_{\alpha}$ is $L^\star$-Lipschitz, but it should be $L_{\star}$. Please unify the notation throughout the manuscript.

**Other Strengths And Weaknesses:**

See the following "Other Strengths And Weaknesses" part.

**Questions For Authors:**

See the "other comments or suggestions" part for details.

**Relation To Broader Scientific Literature:**

In theory part, i.e., Appendix B, the author cite the results in [Curi et al., 2020]. For Theorem B.4 and B.5, I think the proof techniques used are standard, I will add more comments in the following "Other Strengths And Weaknesses" part.

**Theoretical Claims:**

Yes, I check their Appendix B, see the following "Other Strengths And Weaknesses" part for details.

---

> ### Author Rebuttal · Authors · 2025-04-01
>
> Thanks for your positive and valuable feedback. We have made efforts to address your concerns. If there are any further questions, please let us know and we will reply promptly.
>
> **Q1.**   **The contribution seems incremental.**
>
> > **Reply**: While CVaR optimization is well-established (e.g., in [Curi et al., 2020]), we believe our contribution lies in adapting it specifically to the Neural Process (NP) framework. This is non-trivial because tasks in meta-learning differ greatly in difficulty and distribution, making CVaR-based reweighting particularly relevant to “worst-case adaptation.”
> >
> > We provide a finite-sum DRO formulation for task-level NP optimization and find that standard SGD-based solutions for CVaR can suffer from exploding/vanishing gradients. The double-loop mirror prox with variance reduction addresses these problems, stabilizing updates while achieving robust generalization.
> >
> > As with many deep-learning contexts, providing rigorous global convergence guarantees in non-convex scenarios is challenging. We plan to investigate approximate convergence or local convergence guarantees in a future revision, building upon the convex analyses presented in Appendix B.
> >
> >*[Curi et al., 2020] Curi, S., Levy, K. Y., Jegelka, S., and Krause, A. Adaptive sampling for stochastic risk-averse learning. Advances in Neural Information Processing Systems, 33:1036–1047, 2020.*
>
> **Q2.**   **(a) What is the sample complexity and minimum number of iterations required to achieve a certain level of accuracy?(b) In Algorithm 1 (Line 5), how is the maximum number of inner loop iterations Ks determined?**
>
> > **Reply**: We follow standard distributionally robust optimization settings, where each task can be large, and the meta-dataset comprises many tasks. In each iteration, we typically sample at least one data point per task (one shot per task). Detailed formal sample complexity results exist for convex mirror-prox setups; translating them to non-convex NPs remains an open question.
> >
> > In the paper, $S$ denotes the number of outer loops (epochs), and $K_s$ represents the number of inner-loop iterations within each outer-loop $s$. In experiments, we set $K_s$ as a constant across all outer loops, i.e., $K_s = K$ for all $s$. $K$ is chosen to scale with the average number of samples per task. Empirically, we find that $K_s = 100-500$ often suffice for stable training in NP tasks; deeper loops help slightly but yield diminishing returns relative to the added cost. For outer loop epochs $S$, usually between 50 to 500 epochs. For simple tasks (e.g., low-dimensional regression, small datasets),  $S$ is 50–200 epochs.  For complex tasks (e.g., image completion),  $S$ often requires 200–500+ epochs. We will expand them to also show the effect of varying the number of inner/outer loop iterations in a revised version.
>
> **Q3.**   **Theoretical Questions (Appendix B)**
>
> > **Reply**: For Proposition B.1, we will clarify that H refers to a set of finite cardinality or finite VC dimension to avoid ambiguity.
> >
> > For Theorem B.5, the $a$ in the norm $|\cdot|_{\theta a,*} $  is a typo, it should be $|\cdot|_{\theta,*} $.
> >
> > For Line 740 (Eq.(26)), the second term should be $2(\sum_{m=1}^M|q_m^+ - q_m^-|G)^2$. For the second term in line 737, we also omitted the square. Thank you again for pointing out the error.
> >
> > For the left-hand side of Equation (27), adding or omitting the absolute value or norm is acceptable.
> >
> > For Equation (28), we will correct $||\theta^+ - \theta^-||_{\theta}$ to be squared, i.e., $||\theta^+ - \theta^-||_{\theta}^2$
>
> **Q4.**   **Clarifications and Notation Issues**
>
> > **Reply**: Line 127 (Left Column): Garnelo et al. (2018a) assume the target set $T$ is disjoint from the context set $C$ in standard CNP. In our exposition, $T$ could overlap with $C$ or not. The choice is optional and widely used in BNP, DIVNP, TNP, SNP, and we will clarify that it differs slightly from the original CNP assumptions.
> > Lines 220–221: In the expression $max_{\theta} \psi_{\theta}(\theta)−min_{\theta} \psi_{\theta}(\theta) ≤ D_{\theta}^2$, we will revise the notation to separate the parameters inside $\psi(\cdot)$ from the outer $\theta$ symbol, ensuring clarity.
>
> **Q5.**   **Lines 436–437 suggest robust solutions shift the risk distribution left (Figure 4), but in Figure 4(b), RANP still has a rightmost bar. Could you clarify?**
>
> > **Reply**: RANP does not eliminate all high-risk tasks but significantly reduces their proportion. We will refine the text to specify “we shift the distribution, decreasing the fraction of tasks with high risk” rather than implying complete removal. We can also adjust Figure 4’s bin widths or annotate it more explicitly to highlight the reduction percentage.
>
> **Q6.**   **Typographical Errors**
>
> > **Reply**: We have thoroughly revised the spelling errors, typographical mistakes, and symbol ambiguities in the text.
>
> Lastly, thank you once again for your valuable comments.

---

### Official Review · Reviewer_x3vk · 2025-03-22

**Overall Recommendation:** 4

**Summary:**

This paper introduces a new framework for improving the robustness of Neural Processes. Traditional NPs optimize for average performance across tasks using empirical risk minimization, but this can lead to poor adaptation on difficult or high-risk tasks.
The authors propose a risk-averse optimization strategy based on Conditional Value-at-Risk (CVaR), which shifts focus toward minimizing the loss in the worst-performing fraction of tasks (top % highest-risk tasks).

To make CVaR optimization tractable for NPs, the authors reformulate the objective as a finite-sum minimax problem and solve it using a variance-reduced stochastic mirror prox algorithm. This method uses a double-loop structure, where an outer loop computes stable "snapshot" gradients and an inner loop performs refined updates using stochastic, variance-reduced gradients across tasks.

The authors evaluate their method across multiple domains—image completion (CelebA, EMNIST), Bayesian optimization, and contextual bandits; comparing robust NPs (RNPs) against standard, bootstrapped, and stabilized versions of NPs. Across all benchmarks, RNPs consistently demonstrate improved robustness, particularly under data distribution shifts, model-data mismatch, and adversarial task setups. The method effectively reshapes the task risk distribution, reducing the frequency of high-risk task failures.

Overall, the paper contributes a theoretically grounded, practically effective approach for training Neural Processes to be more reliable under task uncertainty and variability.

**Claims And Evidence:**

Most of the paper's key contributions — particularly the need for CVaR-based optimization, the effectiveness of the proposed algorithm, and performance improvements in multiple domains — are supported with clear and convincing evidence.

However, a few theoretical claims would benefit from stronger empirical or quantitative backing:
 - generality: experiments only evaluate a fixed set of standard NP architectures (CNP, ANP, ConvNP, etc.). It’s unclear how well it transfers to significantly different variants or tasks not tested. This is potentially true.
-  bias reduction: This is stated theoretically but not quantitatively evaluated in experiments (no analysis of bias vs. variance trade-off or formal bias reduction metrics).

**Essential References Not Discussed:**

Citations are well written.

**Experimental Designs Or Analyses:**

Minor issues:
---
- Hyperparameter sensitivity: no detailed discussion of sensitivity to CVaR level α, learning rates, or inner/outer loop lengths — which could affect robustness.

- Runtime: no analysis of training time, convergence speed, or compute overhead from the double-loop structure, which would help assess trade-offs.

**Methods And Evaluation Criteria:**

Yes.the proposed methods and evaluation criteria are generally well-aligned with the problem of improving robustness in Neural Processes under task variability and risk

**Other Comments Or Suggestions:**

typos
---
section 3.1: fundom -> random

**Other Strengths And Weaknesses:**

Strengths:
---
- Originality: While the core components (CVaR, Mirror Prox, variance reduction) are established individually, their integration into the Neural Process framework is novel and well-motivated.

- Significance: The proposed method addresses a real and underexplored challenge; robustness to high-risk task failure in few-shot and meta-learning settings. Given the increasing deployment of meta-learning models in uncertain environments (e.g recommendation or robotics), this contribution is both timely and impactful.

- Clarity of technical sections: The paper does a solid job presenting technically dense ideas (e.g., minimax reformulation, Bregman setups) in a logically structured way. The supplementary material is especially helpful, providing detailed derivations and implementation notes.

- Empirical breadth: The experiments span generative modeling, black-box optimization, and decision-making ; showcasing the generality of the approach. The consistent performance improvements across these settings strengthen the empirical case.

Weaknesses:
---
- Clarity for broader audiences: While technically sound, some sections (e.g., optimization formulation and variance reduction) are dense and could benefit from more intuitive explanations or diagrams in the main text.

- Ablation/Interpretability: There is limited ablation on the effect of CVaR level α, inner/outer loop iterations, or mirror prox updates. Understanding the sensitivity of the model to these choices would clarify its practical utility.

- Efficiency analysis missing: The method introduces nontrivial computational overhead (double-loop structure, per-task sampling), but no runtime or convergence speed comparisons are provided. This could be critical for real-world deployment.

- Limited theoretical guarantees: While the optimization method is principled, the paper lacks convergence guarantees in non-convex regimes — a common but notable limitation that could be acknowledged more explicitly.


Overall:
---
The paper presents a significant and novel contribution by adapting risk-averse learning to NPs, backed by thoughtful algorithm design and broad empirical validation.

**Questions For Authors:**

Already addressed in the weaknesses section:

- How sensitive is the performance to the CVaR level α?
- What is the computational overhead of the double-loop optimization compared to standard ERM-trained NPs?

Also,
- It’s not fully explained why improving tail performance doesn't degrade average-case accuracy?

**Relation To Broader Scientific Literature:**

- NPs and Meta-Learning: the paper extends NPs by addressing a critical limitation — their reliance on empirical risk minimization, which can lead to poor performance on outlier tasks. Unlike existing enhancements (e.g., bootstrapped or stabilized NPs), the paper introduces explicit risk-sensitive objectives.

- Risk-averse and robust optimization: the paper is the first to apply CVaR-based risk minimization in NPs, treating task-level adaptation risk as a distribution and focusing on controlling the worst-performing fraction.

- mirror prox and variance reduction: the paper Innovatively adapts these optimization methods to the meta-learning risk-averse process.

**Theoretical Claims:**

The paper does not contain unproven theoretical claims, and where limitations exist (e.g., convergence in non-convex regimes), they are appropriately acknowledged. No significant correctness issues were identified in the stated proofs or derivations.

---

> ### Author Rebuttal · Authors · 2025-04-01
>
> Thanks for your positive and valuable feedback. We have made efforts to address your concerns. If there are any further questions, please let us know, and we will reply promptly.
>
> **Q1.**   **Ablation experiments on CVaR α, inner/outer loop iteration lengths, and learning rate.**
>
> > **Reply**: We appreciate the need for more thorough ablations. Due to space limits, we included a subset of these in Appendix E.3, focusing on α.  Empirically, we have observed that α values in a moderate range (e.g., 0.4–0.6) often strike a good balance between focusing on worst-case tasks and not overly biasing the model toward rare outliers. We show an ablation (Figure 8) demonstrating that too small an α (e.g., 0.1) under-treats high-risk tasks, while an extreme α (e.g., >0.8) can degrade average performance. These observations align intuitively with the idea that CVaR reweights the “tail” portion of tasks.
> >
> > In the paper, $S$ denotes the number of outer loops (epochs), and $K_s$ represents the number of inner-loop iterations within each outer-loop $s$. In experiments, we set $K_s$ as a constant across all outer loops, i.e., $K_s = K$ for all $s$. $K$ is chosen to scale with the average number of samples per task. Empirically, we find that $K_s = 100-500$ often suffice for stable training in NP tasks; deeper loops help slightly but yield diminishing returns relative to the added cost. For outer loop epochs $S$, usually between 50 to 500 epochs. For simple tasks (e.g., low-dimensional regression, small datasets),  $S$ is 50–200 epochs.  For complex tasks (e.g., image completion),  $S$ often requires 200–500+ epochs. We will expand them to also show the effect of varying the number of inner/outer loop iterations in a revised version.
> >
> >We used Adam optimizer with an initial learning rate $5·10^{-4}$ and decayed the learning rate using a cosine annealing scheme.
>
> **Q2.**   **How does the double-loop optimization compare to simpler baselines in computational overhead?**
>
> > **Reply**: The additional overhead mainly comes from (i) maintaining snapshots of gradients and (ii) sampling from each task in each inner iteration. However, because we adopt a task-aware sampling scheme (i.e., one sample per task), the minibatch size remains comparable to that of standard NP training.
> > In practice, we find the extra overhead to be modest—roughly 1.1–1.3× the per-epoch training time compared to standard NPs, depending on the number of tasks and the length of inner loops.
> > We will add more quantitative details in the final version, including a time-per-epoch comparison and possible strategies (e.g., smaller inner loops) to reduce total training time.
>
> **Q3.**   **Why does improving tail performance not degrade average accuracy?**
>
> > **Reply**: Elevated attention to difficult tasks prevents overfitting to “easy” samples and encourages more robust parameter updates. This can reduce overall variance in predictions and in practice, often benefits average-case accuracy. Our experiments (Tables 1, 2, 3) demonstrate that while we significantly reduce high-risk failure rates, the average performance (e.g., log-likelihood) either remains on par or improves slightly compared to standard ERM-based NPs. CVaR reweights tasks in proportion to their risk. Importantly, for moderately chosen α, we are not only training on the worst tasks but rather adjusting the objective so that the upper tail influences the gradient more strongly. The result is a more balanced training signal across all tasks.
>
> **Q4.**   **The paper largely focuses on convex analysis, but deep NPs are inherently non-convex.**
>
> > **Reply**: We agree that in non-convex regimes, strong theoretical convergence guarantees are elusive (an issue for many deep learning algorithms). While mirror-prox methods do enjoy strong convergence rates for convex-concave saddle-point problems, we currently rely on their practical effectiveness for non-convex NPs.
> > We have followed prior work in robust optimization, which often leverages the same theoretical tools in non-convex settings with the understanding that local minima or stable solutions can often suffice in practice. In the final version, we will highlight this limitation more explicitly.
>
> Lastly, thank you once again for your valuable comments.

---

### Decision · Program_Chairs · 2025-05-01

**Decision:**

Accept (poster)

**Comment:**

The reviewers agreed that this paper is well written and  proposes a novel solution to an important problem.
Some reviewers were concerned about the lack of theoretical novelty compared to prior work on CVaR  [Curi et al., 2020], and the lack of  convergence proof for the neural processes setting.

It seems that the rebuttal has addressed most of the concerns raised by the reviewers.

In my opinion, while the theoretical contribution is indeed limited, the integration of the method to the neural processes setting and the empirical results are significant contributions to the community and merit publication in this venue.